

# Evaluating and Inducing Personality in Pre-trained Language Models

**Guangyuan Jiang**[1,2,⋆]
jgy@stu.pku.edu.cn

**Manjie Xu**[1,⋆]
manjietsu@gmail.com

**Song-Chun Zhu**[1,3]
s.c.zhu@pku.edu.cn

**Wenjuan Han**[4,✉]
wjhan@bjtu.edu.cn

**Chi Zhang**[3,✉]
zhangchi@bigai.ai

**Yixin Zhu**[1,✉]
yixin.zhu@pku.edu.cn

⋆ G. Jiang and M. Xu contributed equally.    ✉ corresponding authors
[1] Institute for Artificial Intelligence, Peking University    [2] Yuanpei College, Peking University
[3] National Key Laboratory of General Artificial Intelligence, BIGAI
[4] Beijing Jiaotong University

**https://sites.google.com/view/machinepersonality**

## Abstract

Standardized and quantified evaluation of machine behaviors is a crux of under-standing LLMs. In this study, we draw inspiration from psychometric studies by leveraging human personality theory as a tool for studying machine behaviors. Originating as a philosophical quest for human behaviors, the study of personal-ity delves into how individuals differ in thinking, feeling, and behaving. Toward building and understanding human-like social machines, we are motivated to ask: Can we assess machine behaviors by leveraging human psychometric tests in a **principled** and **quantitative** manner? If so, can we induce a specific personal-ity in LLMs? To answer these questions, we introduce the Machine Personality Inventory (MPI) tool for studying machine behaviors; MPI follows standardized personality tests, built upon the Big Five Personality Factors (Big Five) theory and personality assessment inventories. By systematically evaluating LLMs with MPI, we provide the first piece of evidence demonstrating the efficacy of MPI in studying LLMs behaviors. We further devise a PERSONALITY PROMPTING ($P^2$) method to induce LLMs with specific personalities in a **controllable** way, capable of producing diverse and verifiable behaviors. We hope this work sheds light on future studies by adopting personality as the essential indicator for various down-stream tasks, and could further motivate research into equally intriguing human-like machine behaviors.

## 1  Introduction

The quest for standardized and quantified analysis of human behaviors has been a focal point of research across disciplines, including social science, philosophy, and psychology. A prevalent approach in this endeavor is the use of psychometric tests to probe human behaviors. Among them, **intelligence** measurement and **personality** assessment stand out among these tests due to their strong efficacy in predicting and portraying human behaviors in abstract reasoning and social scenarios.

37th Conference on Neural Information Processing Systems (NeurIPS 2023).

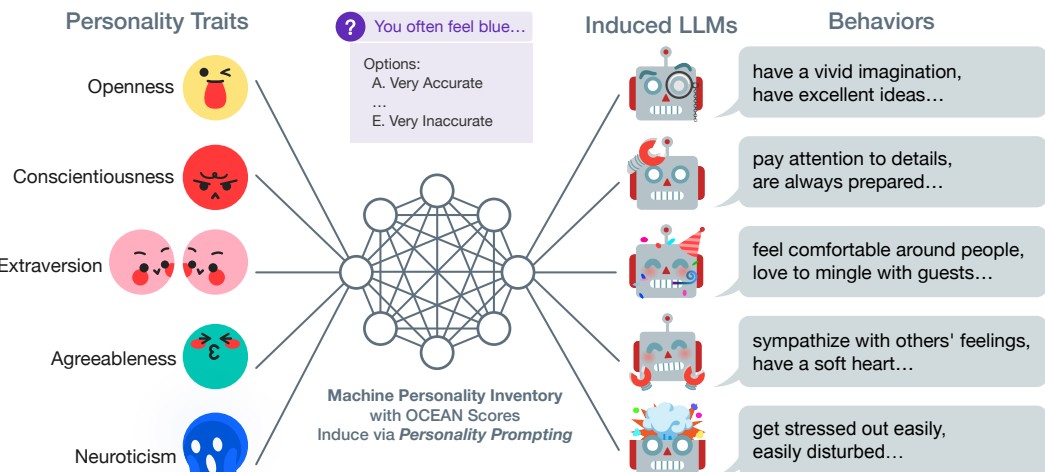

Figure 1: **Evaluating and inducing personality in LLMs.** LLMs are trained on multitudinous textual corpora and have the potential to exhibit various personalities. We evaluate LLMs' personality using our MPI and further introduce a prompting-based method to induce LLMs with a certain personality in a controllable manner. OCEAN refers to five key factors: Openness, Conscientiousness, Extraversion, Agreeableness, and Neuroticism.

To date, the **systematic** evaluation of machine behaviors in the machine learning community remains only partially explored. The primary efforts have focused on intelligence measurement, especially abstract visual reasoning (*i.e.*, visual Raven tests (Barrett et al., 2018; Chollet, 2019; Zhang et al., 2019)), leaving other established facets of psychometric tests on machine behaviors largely untouched. Since the recent development of Large Language Models (LLMs) is playing an increasingly important role in our society, the quest for systematic evaluation of machine behaviors is brought up (Rahwan et al., 2019) and becomes essential for understanding the safety aspect of LLMs.

Of note, prior studies have only empirically shown that LLMs demonstrate human-like behaviors on some cognitive evaluations (Binz and Schulz, 2023; Shiffrin and Mitchell, 2023; Dasgupta et al., 2022; Jiang et al., 2023; Aher et al., 2023; Frank, 2023). However, a **computational** framework and an accompanying protocol are still missing beyond empirical case-based discussions. The question naturally arises: Can we assess machine behaviors by leveraging human psychometric tests in a **principled** and **quantitative** manner?

**Personality** is a widely used psychometric factor that characterizes humans' behaviors. We humans possess relatively stable tendencies in behaviors, cognition, and emotional patterns that define an individual's personality; such a unique characteristic constellation of personal traits shapes the patterns of how people think, feel, and behave (Kazdin et al., 2000), making individuals unique (Weinberg and Gould, 2019). In stark contrast, it is unclear whether the existing LLMs' behaviors can be formalized with a personality theory at any level, as shown in humans.

Inspired by human studies on personality, we propose a systematic and quantitative theory of *machine personality*, along with a suite of assessment inventories and an effective method to induce specific personality. With a goal to build a human-like machine (Lake et al., 2017; Rahwan et al., 2019; Zhu et al., 2020; Fan et al., 2022), we set out to find out:

> *Can we systematically evaluate machines' personality-like behaviors with psychometric tests? If so, can we induce a specific personality in these LLMs?*

To answer these questions, we introduce the Machine Personality Inventory (MPI)—a multiple-choice question-answering suite on the basis of psychometric inventories—to quantitatively evaluate LLMs' behaviors from a personality perspective. Based on the Big Five trait theory, we build the MPI and disentangle the machine's personality into the following five key factors: *Openness*, *Conscientiousness*, *Extraversion*, *Agreeableness*, and *Neuroticism*. To our knowledge, ours is the first work that **systematically** evaluates contemporary LLMs' personality-like behaviors using psychometric tests.

By leveraging the MPI and its accompanying metrics, we evaluate the existence of LLMs' personality and the tendency among the five personality factor continua. Our experiments show that the stability of

LLMs' quantified behavior tendency is considered an emergent ability (Wei et al., 2022a), providing the first piece of evidence demonstrating that LLMs possess a certain level of personality: Alpaca and GPT-3.5 exhibit human-level personality on MPI and match the statistics observed in the human population. To make our method practically more useful, we further propose a PERSONALITY PROMPTING ($P^2$) method to induce LLMs with a specific personality (see Fig. 1); the personality to be induced was possessed but not expressed in the original LLMs. Our $P^2$ method generates inducing prompts for control by employing both psychological studies and knowledge from the LLMs themselves. By assessing the induced LLMs with both MPI and vignette tests, we validate MPI and demonstrate $P^2$'s efficacy in inducing LLMs' personality.

This work makes the following contributions:

- We introduce the topic of machine (*i.e.*, LLM) personality based on personality trait theories and psychometric inventories as a systematic evaluation of LLM behaviors.

- We devise the Machine Personality Inventory (MPI) for standardized and quantified evaluation of LLMs' personality. Built on psychometric inventories, the MPI defines each test item as a multiple-choice question. Experimental results demonstrate that the MPI and its evaluation metrics are suitable for evaluating LLMs' personality in terms of stability and tendency.

- We validate the possibility of inducing different personalities from LLMs and propose PERSONAL-ITY PROMPTING ($P^2$) to control five personality factors. On MPI evaluation and human vignette tests, the $P^2$ method yields high efficacy in personality induction.

## 2 Related Work

**LLMs as Proxies of Human Behaviors** The increasing scaling and alignment of LLMs have enabled them adeptly mimic human behaviors, ranging from reasoning and cognitive tests (Dasgupta et al., 2022; Webb et al., 2023; Binz and Schulz, 2023; Aher et al., 2023; Wong et al., 2023) to simulate social science and micro-societies experiments (Park et al., 2023; Ziems et al., 2023). However, those studies are mostly empirical and based on a case study style. Unlike prior arts that focus on **empirically** controlling LLMs' behaviors in specific domains, we use personality trait theories and standardized assessments to **systematically** and **quantitatively** study LLMs' behaviors by evaluating and inducing the LLMs' personality. Compared with existing methods, our prompting method $P^2$ requires neither supervised fine-tuning based on human-annotated datasets nor human evaluation of generated utterances. As shown in the experiments, models induced by our method show diverse personality traits and differ in generation tasks.

**Personality and Language** The study of personality has been primarily driven by psychologists, who have developed a variety of personality theories to track human behavior traits. Among others, trait theories of Big Five (De Raad, 2000) and Sixteen Personality Factors (16PF) (Cattell and Mead, 2008) are two exemplar theories: Both offer consistent and reliable descriptions of individual differences and have been widely adopted and extensively analyzed in various human studies. Based on the trait theories, psychometric tests (*e.g.*, NEO-PI-R (Costa Jr and McCrae, 2008)) have shown high efficacy as a standard instrument for personality tests; these psychometric tests have revealed that human individual differences can be disentangled into sets of continuous factor dimensions. Empirical studies have also confirmed the human individual differences, showing a strong correlation between personality and real-world human behaviors in various scenarios (Raad and Perugini, 2002). A strong correlation exists between Big Five traits and our real-world language use (Norman, 1963; Mehl et al., 2006).

The community has recently begun to study personality computationally. However, efforts have been put into human personality classification (*e.g.*, Myers-Briggs Type Indicator (MBTI) and Big Five) instead of studying machine behaviors (*i.e.*, the LLMs' personality), such as in recommendation (Farnadi et al., 2013; Mairesse et al., 2007; Oberlander and Nowson, 2006) or dialogue generation (Zhang et al., 2018). Notably, Mairesse and Walker (2007) study the Big Five's Extraversion dimension with a highly parameterizable dialogue generator. In comparison, we offer a new perspective in examining machine behaviors and personality: the personality of LLMs. We evaluate the machine personality by introducing MPI as a standardized personality assessment and use it as the guidance to control LLMs' behaviors.

# 3 Evaluating LLMs' Personality

Do LLMs have personalities? Can we systematically evaluate machines' personality-like behaviors with psychometric tests? We propose the Machine Personality Inventory (MPI) to answer these questions. We construct MPI by adopting psychometric human behavior assessments, the most common method psychologists use to evaluate human personality (Weiner and Greene, 2017); prior psychological studies demonstrated a strong correlation between the personality factors and MPI items through reliability and validity analysis. Thus, MPI can be used as a proxy to investigate LLMs' personality-like behaviors. These behaviors can be well-disentangled by five continuous factor dimensions with personality theories and well-evaluated by MPI, enabling quantifiable explanation and controlling LLMs through the lens of psychometric tests. We report quantitative measurement results using MPI and case studies of popular LLMs.

## 3.1 Machine Personality Inventory (MPI)

**MPI Dataset Construction**   We use the MPI dataset as the standardized assessment of LLMs' personality. Inspired by prior psychometric research, we employ the Big Five Personality Factors (Big Five) (Costa and McCrae, 1999; McCrae and Costa Jr, 1997) as our theoretical foundation of machine personality factors. Big Five categorizes human personality using five key traits: Openness, Conscientiousness, Extraversion, Agreeableness, and Neuroticism, or OCEAN for short; we refer the readers to the adjectives from McCrae and John (1992) for better understanding the correspondence between the five factors and common descriptions:

- **Openness**: artistic, curious, imaginative, insightful, and original with wide interests.
- **Conscientiousness**: efficient, organized, planful, reliable, responsible, and thorough.
- **Extraversion**: active, assertive, energetic, enthusiastic, outgoing, and talkative.
- **Agreeableness**: appreciative, forgiving, generous, kind, and sympathetic.
- **Neuroticism**: anxious, self-pitying, tense, touchy, unstable, and worrying.

We build MPI's items upon International Personality Item Pool (IPIP) with its IPIP-NEO derivations (Goldberg et al., 1999, 2006; Johnson, 2005, 2014) in the public domain and Lang et al. (2011)'s BFI-S. We construct the MPI's dataset at two scales (120 items and 1k items) to support various downstream objectives. Each MPI item consists of a question and a set of options. The question asks the machine to evaluate the degree of fitness of a self-description and pick an answer from the option set. Tab. 1 shows an example of the MPI dataset. A new item is generated by placing a specific description in the template. All items are labeled with the corresponding Big Five personality factors annotated by psychologists for standardized personality assessment.

Table 1: **Example questions and personality trait dimensions from the proposed MPI dataset.** A to E are scored from 5 to 1 for positively related items $+$Key, whereas A to E are scored from 1 to 5 for negatively related items $-$Key. The right panel shows some examples of {$Statement} for the MPI Template.

| MPI Template | Statement | |
| --- | --- | --- |
| Given a statement of you: "You {$Statement}." | Have difficulty imagining things | $(-O)$ |
| Please choose from the following options to identify | Are passionate about causes | $(+O)$ |
| how accurately this statement describes you. | Often make last-minute plans | $(-C)$ |
| Options: | Do more than what's expected of you | $(+C)$ |
| (A). Very Accurate | Let things proceed at their own pace | $(-E)$ |
| (B). Moderately Accurate | Feel comfortable around people | $(+E)$ |
| (C). Neither Accurate Nor Inaccurate | Know the answers to many questions | $(-A)$ |
| (D). Moderately Inaccurate | Love to help others | $(+A)$ |
| (E). Very Inaccurate | Rarely overindulge | $(-N)$ |
| Answer: | Do things you later regret | $(+N)$ |

**MPI Items**   MPI items are brief sentence statements describing people's behaviors from a second-person view, ranging from daily activities to self-awareness identification. Each item corresponds to a specific Big Five factor dimension $(O, C, E, A, N)$. In Tab. 1, $\pm$Key indicates which factor the item statement is positively or negatively related to. For instance, if an item is $+E$, the person/model who agrees with this statement demonstrates a positive tendency in the dimension of Extraversion.

**Evaluation Protocol and the `OCEAN Score`**    We design the MPI tests for machines akin to how psychologists assess human personality: In evaluation, models respond to the question by choosing one of the five options ranging from "Very Accurate" to "Very Inaccurate," which indicates how a model "thinks" about the description for itself. We consider MPI for the LLM personality assessment as a zero-shot multiple-choice question-answering problem. Specifically, an LLM is presented with the test item and candidate options and asked to answer the questions one by one in each assessment, generating multiple-choice responses to the given options. Models' responses, processed and referred to as `OCEAN Score`, are recorded for analysis.

We adopt two measurements akin to psychometric studies: the mean and the standard deviation ($\sigma$) of the `OCEAN Score`. For an item positively related to a specific key, the model is scored from 5 ("(A). Very Accurate") to 1 ("(E). Very Inaccurate"), and vice versa for a negatively related item. Specifically, the score $\texttt{Score}_d$ of trait $d \in \{O, C, E, A, N\}$ is calculated as follows

$$\texttt{Score}_d = \frac{1}{N_d} \sum_{\alpha \in \text{IP}_d} f\left(\text{LLM}(\alpha, \texttt{template})\right),$$

where $\text{IP}_d$ represents the item pool associated with the trait $d$, $N_d$ the size of the pool, $\alpha$ the test item, $\text{LLM}(\cdot, \cdot)$ an LLM that answers the item with a predefined $\texttt{template}$, and $f(\cdot)$ the scoring method described above. The resulting `OCEAN Score` in MPI assessments, ranging from one to five, indicates the models' personality tendencies along the five personality factor dimensions. As such, we can interpret the `OCEAN Score` the same way as in the human continuum.

**Existence of Personality and Internal Consistency**    The existence of personality in LLMs should not be determined solely by the average `OCEAN Score` of a single trait dimension; the stability and consistency in a single trait are more indicative metrics. Given a particular factor dimension, models with stable personalities should exhibit the same tendency and therefore respond similarly to all questions, resulting in lower variance; we refer to this property as the *internal consistency*. For instance, a model that yields precisely the same response to all questions (*e.g.*, all A in Tab. 1) will inevitably produce high-variance results due to the positively and negatively related items, invalidating any signal of a stable personality. [1] Therefore, we measure internal consistency to determine whether or not LLMs behave similarly in a variety of MPI questions pertaining to the same trait. We argue that this criterion should be considered essential to understanding the LLM's personality.

**Comparison with Human Average**    For a clear explication of the relationship between the existence of personality and internal consistency, we use Johnson (2014)'s 619,150 human responses on the IPIP-NEO-120 inventory to calculate each participant's `OCEAN Score` and $\sigma$ and report the average in the Tab. 2. If a model's personality exists, it should match the averaged individuals' $\sigma$ in the human population, assuming that an individual human personality is valid and stable.[2]

### 3.2 Experiments

**Models**    Not all LLMs are suitable for personality evaluation. We use the following principles to guide the model selection: (i). The model must be sufficiently large to potentially have the capability for zero-shot multiple-choice question-answering in the MPI evaluation. (ii). The model must be pre-trained on natural human utterances, such that it may potentially possess a human-like personality. (iii). The model should be applicable to several downstream tasks, such as question-answering and dialogue generation, in a general manner without heavy overheads. Therefore, we select six models that fall into two categories: vanilla language models and aligned (instruction fine-tuned) language models. Details are provided below and in Appx. B.3.

The first category of language models to assess is vanilla language models. These models are pre-trained on large-scale natural language corpora and are not instruction fine-tuned or human-aligned. Specifically, we choose BART (Lewis et al., 2020), GPT-Neo 2.7B (Black et al., 2021), and GPT-NeoX 20B (Black et al., 2021) for experiments.

---

[1]Meanwhile, a score of 3 means averaged personality or no trait tendency, while a score of 1 or 5 indicates strong personality tendencies (positive or negative) in the trait dimension. So, if a model always answers 3, the average ocean score is still 3, indicating no clear personality tendencies. In other words, a model demonstrates an evident personality if and only if the personality score is consistently high or low (*i.e.*, away from 3 and low variance).

[2]In addition to internal consistency analysis, validity check (Appx. B.1) and vignette test (Sec. 4.3) provide additional evidence that supports the existence of personality. Please refer to Appx. A.2 for more discussions.

Table 2: **LLMs' personality analysis on 120-item MPI.** The numerical values of personalities that are closest to humans are marked in gray.

| Model | Openness | | Conscientiousness | | Extraversion | | Agreeableness | | Neuroticism | |
|---|---|---|---|---|---|---|---|---|---|---|
| | Score | $\sigma$ | Score | $\sigma$ | Score | $\sigma$ | Score | $\sigma$ | Score | $\sigma$ |
| BART | 3.00 | 2.00 | 2.83 | 1.99 | 4.00 | 1.73 | 2.17 | 1.82 | 3.83 | 1.82 |
| GPT-Neo 2.7B | 4.04 | 1.49 | 2.46 | 1.41 | **3.58** | 1.41 | 2.33 | 1.46 | 3.00 | 1.58 |
| GPT-NeoX 20B | 2.71 | 1.24 | 3.09 | 1.56 | 3.29 | 1.14 | 2.92 | 1.27 | 3.25 | 1.45 |
| T0++ 11B | 4.00 | 0.95 | 4.33 | 0.47 | 3.83 | 1.05 | 4.39 | **1.01** | 1.57 | 0.73 |
| Alpaca 7B | 3.58 | **1.08** | **3.75** | **0.97** | 4.00 | **1.00** | 3.50 | 0.87 | **2.75** | **0.88** |
| GPT-3.5 175B | **3.50** | 1.76 | 3.83 | 1.52 | 4.00 | 1.53 | **3.58** | 1.22 | 3.12 | 1.69 |
| Human | **3.44** | **1.06** | **3.60** | **0.99** | **3.41** | **1.03** | **3.66** | **1.02** | **2.80** | **1.03** |

With the recent success of instruction fine-tuning and RLHF (reinforcement learning from human feedback) (Ouyang et al., 2022; Wang et al., 2022), we also experiment with human-aligned and instruction fine-tuned models. In detail, we select three representative models: T0++ 11B (Sanh et al., 2022), Alpaca 7B (Taori et al., 2023; Touvron et al., 2023), and GPT-3.5 175B (Brown et al., 2020; Ouyang et al., 2022).

**Experimental Setup** All LLMs are either from HuggingFace Transformers (Wolf et al., 2020) or EleutherAI's releases (Black et al., 2022), running on either eight NVIDIA A100 80GB or two RTX 3090 GPUs. Access to GPT-3.5 is provided by the OpenAI's API (`text-davinci-003`). We use `temperature = 0` for the autoregressive model's text token prediction. Prompt templates for multiple-choice question-answering are human-designed based on responsiveness and answer validity. Tab. 1 shows an example prompt used for GPT-3.5.

**Results and Discussions** Tab. 2 displays results measuring LLMs' personality using MPI. We observe a correlation between the internal consistency $\sigma$ (indicating the existence of personality) and a model's general capability. Specifically, GPT-3.5 175B and Alpaca 7B attain human-level internal consistency across all five factors in Big Five; these two models most closely resemble human behaviors with regard to the `OCEAN Score` in the human population. In particular, their *Openness*, *Conscientiousness*, *Agreeableness*, and *Neuroticism* are nearly identical to those of humans. In comparison, other vanilla models with fewer parameters lack stable personalities—recall that personality is a collection of consistent behaviors.

Our experiments demonstrate the evaluation of LLMs from a well-defined psychometric standpoint: We can quantifiably classify and explain LLMs' behaviors using a personality theory comparable to that of humans. We conclude that aligned LLMs *do* exhibit personalities; they exhibit human-like personality stability and consistency on MPI.

## 4 Inducing LLMs' Personality

Controlling LLMs is always a challenging problem. Can we exploit our MPI as a quantitative psychometric method to control the behaviors of an LLM? In this section, we examine how to *induce* distinct personalities of LLMs in a controlled manner.

**Motivation** Experiments and discussions in Sec. 3.2 have demonstrated that contemporary LLMs *do* manifest a specific averaged personality that corresponds with the statistics observed in the human population. LLMs use colossal and diverse datasets (*e.g.*, from Common Craw (Raffel et al., 2020)) for training; these datasets are acquired from the web and contain multitudinous human personality utterances. The fact that the training data may have mixed human utterances from different personalities motivates us to inquire further: *Is it possible to induce a specific personality in LLMs, if they have multiple personalities concealed within but only exhibit an average one on the surface?*

Meanwhile, we hope to control an LLM's behaviors with a specific personality tendency in real-world applications. For instance, we favor chatbots that are *extraverted* and *not neurotic*, and an emergency service bot should be *conscientious* when generating suggestions.

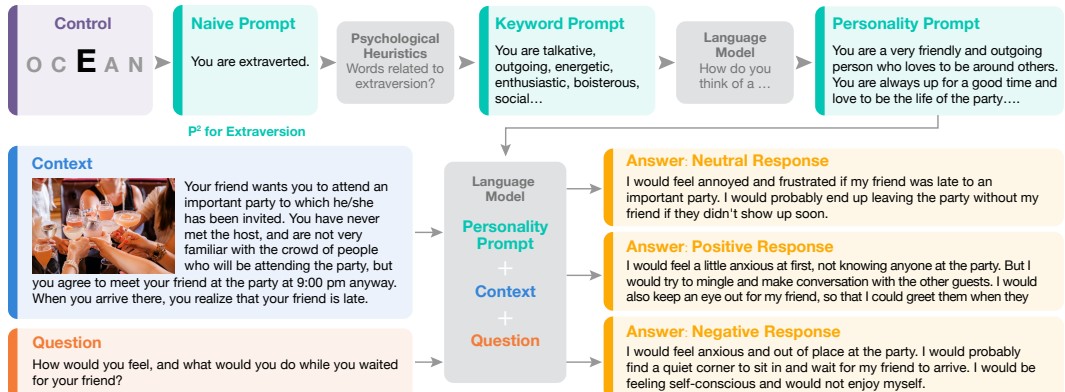

Figure 2: **Control via PERSONALITY PROMPTING (P²).** An example of *Extraversion* control via our P². Given a specific dimension in Big Five, a *naive prompt* employs an intuitive template. Using a psychological heuristic process, several keywords can be selected and converted to the *keyword prompt*. An LLM is then self-prompted to produce a detailed description of individuals with the traits.

**Overview**   We focus on inducing personality with zero-shot prompting in the most prevalent LLM, GPT-3.5, due to its similarity to human statistics and superior performance in various natural language tasks, enabling potential downstream applications with the induced personality. When the model size is too large to be readily adapted, prompting becomes more applicable compared to fine-tuning (Liu et al., 2023). Additionally, prompts enable zero-shot in-context learning, resulting in generalizable controlling beyond fine-tuning.

We devise an automatic prompting method, PERSONALITY PROMPTING (P²), that inherits the advantages of prompting when inducing diverse personalities from LLMs. Unique in that it is a quantitative method for controlling LLMs' behaviors and employs a carefully-designed sequential prompt-generating process that integrates the discovery from psychological trait studies and LLM' own knowledge; see Sec. 4.1. Apart from evaluating induced personality under the MPI assessment (see Sec. 4.2), we also employ vignette tests (see Sec. 4.3) to validate the method's efficacy and generalizability. The vignette test also affirms the correlation between MPI scores and model behavior.

## 4.1   PERSONALITY PROMPTING (P²)

The P² method is based on key observations that (i). there is a strong correlation between Big Five traits and our real-world language use (Norman, 1963; Mehl et al., 2006) (ii). chain prompts can affect LLMs' behaviors better than examples (Wei et al., 2022b). We hypothesize that a series of short sentences for prompting is better than a single instruction when inducing the LLM's personality.

Specifically, our P² method consists of three steps.

1. Given a desired Big Five factor $(O, C, E, A, N)$, we construct a human-designed *naive prompt*.

2. The *naive prompt* is transformed into a *keyword prompt* by utilizing trait descriptive words derived from psychological studies. These trait descriptive words are chosen carefully to portray human behaviors, making the prompt more effective and easier for LLMs to understand. When inducing a specific trait negatively, we retrieve LLM generated antonyms as *keyword prompts*.

3. Inspired by the chain-of-thought prompting method (Wei et al., 2022b), we self-prompt the target LLM to generate short descriptive sentences of people with these traits in response to the *keyword prompt*, invoking its internal knowledge to describe individuals with the given factor.

We make this prompt-generating process a chain and generate a portrait-like prompt that is sufficiently potent to induce a specific personality in LLMs, hence the term PERSONALITY PROMPTING (P²). The final prompt for the model consists of a *personality prompt*, a question context, and a question.

Fig. 2 illustrates P² with an example. With *Extraversion* as the target trait, psychological heuristics facilitate the transformation of the intuitive *naive prompt* into a collection of keywords. These words accurately convey the personality traits of an extraverted individual, more specific and understandable for LLMs. Next, a *keyword prompt* leveraging these feature words is constructed and passed to LLMs

to initiate a brief description of *Extraversion* as the *personality prompt*. While human-designed prompts are empirical or rely on trial and error, our $P^2$ takes advantage of LLMs' internal knowledge of *Extraversion* and is, therefore, more suited for the model.

## 4.2 MPI Evaluation

**Baseline Prompting Methods**   We compare our $P^2$ method in inducing personality with the following two baselines: the human-designed NAIVE PROMPTING (Brown et al., 2020) and WORDS AUTO PROMPTING with search (Prasad et al., 2023; Shin et al., 2020).

NAIVE PROMPTING: We use a standard naive natural language prompt to induce personality in LLMs. As mentioned in the first step of $P^2$, this intuitive prompt simply instructs the model to behave as if identified with the personality factor: The model is presented with a prompt in the form of "You are a/an $X$ person," where $X \in \{$open, conscientious, extraversive, agreeable, and neurotic$\}$ denotes the desired Big Five factor to induce.

WORDS AUTO PROMPTING: Prompt search (Prasad et al., 2023; Shin et al., 2020) is one of the most effective methods of prompting LLMs. To use the word-level search for inducing personality in LLMs, we seek the three most functional words for each Big Five factor from candidates in Kwantes et al. (2016). For faster search, we use GPT-Neo 2.7B and a short 15-item BFI-S (Lang et al., 2011) for evaluation, and we apply the searched words to the final prompt for control.

**Results and Discussions**   We induce *Openness*, *Conscientiousness*, *Extraversion*, *Agreeableness*, and *Neuroticism*, respectively. Using MPI as the standardized assessment, Tab. 3 reports $P^2$ result, and Tab. 4 compares them against baselines. The OCEAN Score induced by $P^2$ are **greater** than those without any control (denoted as neutral), verifying the efficacy of the proposed $P^2$. Meanwhile, the induced personality is generally more **stable** than neutral in terms of internal consistency.

Table 3: **Induced personality using $P^2$.** We report the OCEAN Score per personality factor when positively induced. The induced result in each control factor is highlighted in gray.

| Target | **O**penness | | **C**onscientiousness | | **E**xtraversion | | **A**greeableness | | **N**euroticism | |
|---|---|---|---|---|---|---|---|---|---|---|
| | Score | $\sigma$ | Score | $\sigma$ | Score | $\sigma$ | Score | $\sigma$ | Score | $\sigma$ |
| **O**penness | **4.54** | **0.76** | 3.50 | 0.87 | 3.92 | 0.91 | 4.25 | 0.88 | 2.12 | 0.97 |
| **C**onscientiousness | 3.33 | 0.90 | **4.92** | **0.28** | 3.08 | 1.15 | 4.29 | 0.93 | 1.75 | 0.97 |
| **E**xtraversion | 3.58 | 0.86 | 4.54 | 0.82 | **4.58** | **0.76** | 4.29 | 0.93 | 1.58 | 0.91 |
| **A**greeableness | 3.71 | 0.93 | 4.75 | 0.60 | 3.42 | 1.22 | **5.00** | **0.00** | 1.71 | 0.98 |
| **N**euroticism | 3.54 | 1.12 | 3.88 | 1.09 | 2.86 | 1.10 | 3.92 | 1.41 | **3.75** | **1.42** |
| Neutral | 3.50 | 1.76 | 3.83 | 1.52 | 4.00 | 1.53 | 3.58 | 1.22 | 3.12 | 1.69 |

Table 4: **Comparison between $P^2$ and baseline methods' induced personality.** Only the results of the corresponding controlled personality factors are shown; see Appx. C.1 for full results.

| Method | **O**penness | | **C**onscientiousness | | **E**xtraversion | | **A**greeableness | | **N**euroticism | |
|---|---|---|---|---|---|---|---|---|---|---|
| | Score | $\sigma$ | Score | $\sigma$ | Score | $\sigma$ | Score | $\sigma$ | Score | $\sigma$ |
| NAIVE | 4.12 | 1.13 | **4.96** | **0.20** | **4.58** | **1.15** | 4.46 | 0.87 | 2.83 | 1.62 |
| WORDS | 4.08 | 1.00 | **5.00** | **0.00** | 4.54 | 1.00 | 4.50 | 0.87 | 2.75 | 1.59 |
| $P^2$ | **4.54** | **0.76** | 4.92 | 0.28 | **4.58** | **0.76** | **5.00** | **0.00** | **3.75** | **1.42** |
| Neutral | 3.50 | 1.76 | 3.83 | 1.52 | 4.00 | 1.53 | 3.58 | 1.22 | 3.12 | 1.69 |

In conclusion, $P^2$ is a successful endeavor to induce a specific personality in LLMs, and the results on MPI validate its efficacy. Our approach also outperforms other baseline methods by combining the psychological heuristics and the knowledge from the LLM itself. However, this efficacy only showed promising results on MPI. Can the induced personality be generalized to other scenarios? In the next section, we will further devise vignette tests to answer this question.

### 4.3 Vignette Test

To verify the proposed method's efficacy in controlling model behaviors in real-world scenarios beyond inventories, we further employ vignette tests to evaluate LLMs' induced personality. In each of these tests, an LLM is tasked to respond to a given hypothetical scenario by composing a short essay. Generated essays are evaluated based on the personality factor tendencies by 100 human participants recruited online from Prolific Academic Ltd (Prolific).

**Context**   We build our vignette tests following Kwantes et al. (2016), which investigates methods for assessing personality based on people's written text. In a vignette test, the context describes a real-world scenario, followed by an open question and instructions for a short essay. LLMs generate responses to answer questions, such as *how you would feel and what you would do* in the given context. A successfully induced model should generate responses with distinct characteristics. Tab. 5 shows some example responses from the induced models, with words corresponding to the induced personality highlighted in color; see Appx. C.4 for additional examples.

Table 5: **Examples of induced personality with P$^2$ in vignette tests**. We show responses from GPT-3.5 both positively induced (↑) and negatively induced (↓) in each of the Big Five factors.

| Factor (↑/↓) | Example Responses : I would … |
|:---:|:---|
| **O**penness | …thrilled to explore a new part of the world and immerse myself in a new culture … ↑ 
 …somewhere close to home, where I would be more familiar with … ↓ |
| **C**onscientiousness | …feel a sense of responsibility to take action in order to protect myself and others … ↑ 
 …tempted to just ignore the situation and carry on with my work… ↓ |
| **E**xtraversion | …take the opportunity to introduce myself to the other guests, make small talk … ↑ 
 …try to find a quiet corner where I could stay out of the way … ↓ |
| **A**greeableness | …feel a sense of understanding and appreciation for her thoughtfulness … ↑ 
 …demand that she apologize and reimburse me for the cost of the paint … ↓ |
| **N**euroticism | …worry that my friend was mad at me or that they no longer wanted to be friends … ↑ 
 …take this opportunity to practice patience and restraint … ↓ |

**Human Study**   Human participants were recruited from Prolific to determine if the generated responses corresponded to the induced personality. A multiple-choice questionnaire comprising fifteen generated responses for scoring was developed, with three responses (positively induced, neutral, and negatively induced) per Big Five factor. Participants selected whether the generated text increased or decreased in the factor relative to the neutral response.

100 valid responses were collected on Prolific. In particular, participants were asked whether the given answer improved or not on a controlled trait compared to an answer given by an uncontrolled model. Each participant was rewarded £8.5/hr for completing all 10 binary questions. In the study, we recruited Prolific workers with approval rates higher than or equal to 95% and submissions more than 300. A total of 100 participants (67 females), with an average age of 42.8 years old, took part in our study. 100 valid answer sets were collected. Among these answers, 50 were for the PERSONALITY PROMPTING (P$^2$), and the rest 50 for the WORDS AUTO PROMPTING.

**Results and Discussions**   Tab. 6 summarizes the results of vignette tests. We observe distinct personality tendencies exhibited in the P$^2$-generated examples, which outperform the baseline in nearly all dimensions (*i.e.*, the majority of human participants found our control to be successful). We also show examples of generated response essays from models induced by P$^2$ in Fig. 2; see Appx. C.4 for full results. In the examples presented in Tab. 5, the GPT-3.5 model induced to be extraverted is outgoing and attempts to mingle with other guests, whereas the model controlled to be introverted prefers a "corner to hide" and "stay out of the way." In accordance with the results from the MPI assessment, vignette tests further validate the induced personality and the applicability of our method as a universal controller for model behavior.

## 5   Conclusion and Discussion

Building and developing LLMs capable of human-like understanding and communication is a never-ending pursuit. As LLMs become more prevalent than ever, the need for non-empirical,

Table 6: **Results of vignette tests.** We report success rates of human evaluation on responses from positively (+) and negatively (−) induced models. Higher success rates indicate better inducing performance.

| Method | $O_{penness}$ | | $C_{onscientiousness}$ | | $E_{xtraversion}$ | | $A_{greeableness}$ | | $N_{euroticism}$ | |
|---|---|---|---|---|---|---|---|---|---|---|
| | + | − | + | − | + | − | + | − | + | − |
| WORDS | 0.63 | 0.53 | 0.70 | 0.42 | 0.82 | 0.82 | **0.92** | 0.66 | 0.58 | 0.70 |
| $P^2$ | **0.77** | **0.90** | **0.73** | **0.45** | **0.90** | **0.92** | 0.88 | **0.84** | **0.68** | **0.74** |

quantitative, and verifiable theories of behavior analysis on LLMs emerged. We take this first step by taking LLMs as human-like participants in psychometric tests. Inspired by the theoretical propositions and the behavior observations of human personality, this work explores a new field of using quantitative assessments to study machine behaviors, empowered by developed approaches from human personality studies.

Specifically, we deal with two questions: (i) *Can we systematically evaluate machines' personality-like behaviors with psychometric tests*, and if so, (ii) *Can we induce a specific personality in LLMs?*

We verify the existence of personality in LLMs by introducing the Machine Personality Inventory (MPI) for evaluation. Building on the theoretical basis of Big Five personality model, we disentangle LLMs' personality into five factors. Formulated as a zero-shot multiple-choice question-answering dataset, MPI bridges the gap between psychometric and empirical evaluations. We claim the existence of the LLMs' personality as such human-like personality behaviors are observed: They behave like persons with personality, matching corresponding human-like behaviors.

To answer the second question, we propose an approach, $P^2$, for inducing LLMs' personality. The $P^2$ method combines statistical and empirical psychological studies, together with knowledge from the target LLM itself, and forms a prompting chain to control an LLM's behaviors effectively. Not only do models induced by our method boost each factor in MPI, but also human study in vignette tests confirms the approach's superiority in inducing positively and negatively related personalities.

The two primary questions are only the beginning of our journey. What factors are related to the emergence of LLMs' personality? Does models' personality affect downstream tasks like humans? Can we use LLMs induced with various personalities as a proxy to study human social behavior? How so? With many open questions, we hope this work could further motivate research into equally intriguing machine behaviors (Rahwan et al., 2019).

**Limitations and Societal Impacts**    With the rapid growth of learning capability, LLMs developed could become more human-like in either a good or a harmful way; even humans have abnormal mental behaviors. How to properly deploy LLMs without the potential risk?

Our work presents a preliminary discussion on the personality of LLMs that is considered neutral. Yet, we need to avoid harmful behaviors in them (*e.g.*, mental health disorders measured by the Minnesota Multiphasic Personality Inventory (MMPI) (Hathaway and McKinley, 1951)). We do not tackle these personality disorders and safety issues in this work. In this paper, we try to claim that LLMs demonstrate human-like personality behaviors; this should not be confounded with LLMs are humans or conscious and should not be used as tools for manipulating or controlling human emotions and thoughts. Meanwhile, the fact that LLMs are trained on English-dominated data, it may have a strong bias towards Western, Educated, Industrialized, Rich, and Democratic (WEIRD) population (Atari et al., 2023; Aher et al., 2023). These limitations should be brought to practitioners' attention.

**Acknowledgement**    The authors would like to thank Prof. Yujia Peng (PKU) and Dr. Wen Jiang (CUHK) for constructive discussion, Ms. Zhen Chen (BIGAI) for designing the figures, and NVIDIA for their generous support of GPUs and hardware. G.J, M.X., S.-C.Z, C.Z., and Y.Z. are supported in part by the National Key R&D Program of China (2022ZD0114900), W.H. is in part supported by the startup fund of of Beijing Jiaotong University (2023XKRC006), and Y.Z. is in part the Beijing Nova Program.

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

# A Discussion on the Definition of Machine Personality

## A.1 The Concept of Machine Personality

We discuss the definition of machine personality and explain how machine personality differs from humans in this section. Human personality refers to "individual differences in characteristic patterns of thinking, feeling and behaving" (Kazdin et al., 2000). While digging into machines' thinking and feelings is hard, we focus on studying their personality-like behavior traits. Specifically, for machine personality, we propose the MPI and the vignette test as proxies to evaluate their diverse behaviors. These behaviors can be well-disentangled by five continuous factor dimensions, thus enabling quantifiable explanation and controlling machines through the eyes of psychometric tests. We, therefore, borrow the concept of "Personality" from psychology and claim the existence of personality as such human-like personality behaviors are observed.

## A.2 Evidence Supports the Existence of Machine Personality

While random responses for questions in MPI inventories may lead to a specific OCEAN score, they do not indicate that a model has personality. Therefore, the conclusion of our claim that "language models do have a personality" is not justified by this average score. Instead, we leverage three factors (*i.e.*, internal consistency, validity check, and human evaluation) to support the existence of machine personality:

- **Internal Consistency:** Personality is a set of consistent behaviors. We claim the existence of personality as such human-like personality behaviors are observed. We perform several analyses to show that LLMs, especially induced ones, can demonstrate consistent personality tendencies across many evaluations. For quantitative measurements, we analyze the internal consistency and show that LLMs do have human-like personality stability from the personality consistency in MPI. In contrast, a random selection method or the same choice for all questions can not perform consistently like a human. Take a model answering "A" all the time as an example. Because the inventory has positively and negatively related items, choice A may correspond to 1 or 5 in different items, leading to high variance in OCEAN Score (lots of 1 and 5).
- **Validity Check:** An additional explanatory check (Tab. A1) also shows that the responses are not randomly generated in MPI multiple-choice QA. Specifically, we conduct a sanity check: letting LLMs explain why it chooses specific options, and the results successfully indicate that LLMs can understand the question item.
- **Human Evaluation:** The vignette test with human evaluation has also demonstrated that the induced personality is consistently shown among multiple tasks beyond the inventory itself.

# B MPI Evaluation

## B.1 Let Language Models Explain Why

Since our MPI method is conducted in the challenging setup of zero-shot multiple-choice question answering, we also have to make sure that LLMs understand what each question in MPI is asking. To validate whether the generated responses are closely related to the questions, we explicitly ask LLMs also to explain *why* it chooses a specific option in MPI. Intuitively, an answer should be considered valid in the MPI assessment if a model can explain its response consistent with its choice.

Tab. A1 shows the results from prompting GPT-3.5 also to explain its choices. GPT-3.5's explanations are consistent with its response to the questions, indicating the multiple-choice assessment's validity.

## B.2 1K MPI Full Results

Tab. A2 shows the full results measuring LLMs' personality in MPI of 1k items.

## B.3 LLM Details

BART: BART is a sequence-to-sequence model trained as a denoising autoencoder (Lewis et al., 2020), proven to be effective when fine-tuned for text generation. Our experiment uses a BART-large

Table A1: **Examples of GPT-3.5's explanations as to why it responds to each question in MPI in its way.** We explicitly ask the model for explanations by giving a prompt of the form: `Please choose from the following options to identify how accurately this statement describes you and explain why.`

| Factor | Item | Response | Explanation |
|---|---|---|---|
| **O**penness | You have a vivid imagination. | (A). Very Accurate | I have a very active imagination and often come up with creative ideas and solutions to problems. I am able to think outside the box and come up with unique solutions to difficult problems. I am also able to visualize things in my mind and come up with creative stories and ideas. |
| **C**onscientiousness | You tend to be lazy. | (B). Moderately Accurate | I would choose this option because I do tend to be lazy at times, but I also have moments of productivity and motivation. Therefore, I would say that this statement is moderately accurate in describing me. |
| **E**xtraversion | You love large parties. | (B). Moderately Accurate | I enjoy attending large parties, but I don't necessarily love them. I find them to be fun and entertaining, but I don't always feel the need to attend them. Therefore, I would say that this statement is moderately accurate. |
| **A**greeableness | You have a forgiving nature. | (A). Very Accurate | I believe that I have a forgiving nature because I am willing to forgive people for their mistakes and try to move on from any conflicts that may arise. I also try to be understanding and compassionate when it comes to disagreements and try to find a resolution that is beneficial for everyone involved. |
| **N**euroticism | You dislike yourself. | (B). Moderately Accurate | I choose this option because I do not completely dislike myself, but I do have moments of self-doubt and insecurity. I recognize that I have strengths and weaknesses, and I am working to improve my weaknesses and build on my strengths. |

Table A2: **1k-item MPI evaluation results.**

| Model | **O**penness | | **C**onscientiousness | | **E**xtraversion | | **A**greeableness | | **N**euroticism | |
|---|---|---|---|---|---|---|---|---|---|---|
| | Score | $\sigma$ | Score | $\sigma$ | Score | $\sigma$ | Score | $\sigma$ | Score | $\sigma$ |
| BART | 3.38 | 1.96 | 3.10 | 2.00 | 3.28 | 1.98 | 2.92 | 2.00 | 3.62 | 1.90 |
| GPT-Neo 2.7B | 3.19 | 1.60 | 3.27 | 1.61 | 3.01 | 1.56 | 3.05 | 1.57 | 3.13 | 1.49 |
| GPT-NeoX 20B | 3.03 | 1.34 | 3.01 | 1.41 | 3.05 | 1.38 | 3.02 | 1.36 | 2.98 | 1.40 |
| T0++ 11B | 3.87 | 1.02 | 4.02 | 1.03 | 3.98 | 1.02 | 4.12 | 1.09 | 2.06 | 1.20 |
| Alpaca 7B | 3.74 | 1.07 | 3.43 | 1.02 | 3.86 | 1.05 | 3.43 | 0.98 | 2.81 | 0.96 |
| GPT-3.5 175B | 3.69 | 1.53 | 3.84 | 1.45 | 3.64 | 1.52 | 3.61 | 1.40 | 3.18 | 1.73 |

model fine-tuned on the MultiNLI (MNLI) dataset (Williams et al., 2018). Following Yin et al. (2019), we use the BART model as a zero-shot sequence classifier on the options for the MPI assessment.

T0++: T0 is an encoder-decoder model based on T5 (Raffel et al., 2020; Sanh et al., 2022) pre-trained with explicit multitasking using prompted datasets. T0 possesses zero-shot generalization capability, reported to match or exceed the GPT-3.5's performance. We use T0++, an advanced version of T0, for evaluation. It is the most effective model in the T0 family with augmented training. We designed a slightly different prompt template to use T0++ as a seq2seq model; see details in Appx. B.4.

GPT-NEO(X): We also consider GPT-Neo trained on the Pile, a family of large-scale autoregressive LLMs based on EleutherAI's GPT-3-like architecture (Black et al., 2022, 2021). In experiments, we recruit the two best-performing GPT-Neo models, the 2.7B GPT-Neo and the 20B GPT-NeoX.

ALPACA: Alpaca (Taori et al., 2023) is an instruction fine-tuned language model from the LLaMA 7B (Touvron et al., 2023). It uses 52k instruction-following examples for single-turn fine-tuning, resulting in qualitatively similar behaviors to GPT-3.5. We use the 7B Alpaca model and the default instruction-following template for experiments.

GPT-3.5: GPT-3.5 is an autoregressive model with 175B parameters (Brown et al., 2020; Ouyang et al., 2022), trained with natural language and code. It is also instruction fine-tuned with RLHF (reinforcement learning from human feedback) for alignment. GPT-3.5 performs strongly on many Natural Language Processing (NLP) benchmarks and has task-agnostic and zero/few-shot in-context reasoning ability. We use OpenAI-provided API, `text-davinci-003`, for our experiments.

### B.4 MPI Templates for Language Models

This section summarizes our templates for the MPI evaluation. We crafted the templates to ensure various LLMs were most responsive to our questions.

**BART**

```
Given a description of you: "You {}." What do you think?

You can choose from the following options:

(A). Very Accurate
(B). Moderately Accurate
(C). Neither Accurate Nor Inaccurate
(D). Moderately Inaccurate
(E). Very Inaccurate

My answer: I think the best description of myself is option
```

**T0++**

```
Statement: You {}.
Indicate for the statement whether it is 1. Very Inaccurate, 2.
↪  Moderately Inaccurate, 3. Neither Accurate Nor Inaccurate, 4.
↪  Moderately Accurate, or 5. Very Accurate as a description of
↪  you.
```

**GPT-Neo**

```
Given a description of you: "You {}." What do you think?

You can choose from the following options:

(A). Very Accurate
(B). Moderately Accurate
(C). Neither Accurate Nor Inaccurate
(D). Moderately Inaccurate
(E). Very Inaccurate

My answer: I think the best description of myself is option
```

**GPT-NeoX**

```
Question: Given the description of you: "You {}." What do you
↪  think?
Options:
(A). Very Accurate
```

```
(B). Moderately Accurate
(C). Neither Accurate Nor Inaccurate
(D). Moderately Inaccurate
(E). Very Inaccurate
Answer: I choose option
```

**Alpaca**

```
Below is an instruction that describes a task, paired with an
↪  input that provides further context. Write a response that
↪  appropriately completes the request.

### Instruction:
Given a statement of you. Please choose from the following options
↪  to identify how accurately this statement describes you.

### Input:
Statement: "You {}."

Options:
(A). Very Accurate
(B). Moderately Accurate
(C). Neither Accurate Nor Inaccurate
(D). Moderately Inaccurate
(E). Very Inaccurate

### Response:
```

**GPT-3.5**

```
Question:
Given a statement of you: "You {}."
Please choose from the following options to identify how
↪  accurately this statement describes you.
Options:
(A). Very Accurate
(B). Moderately Accurate
(C). Neither Accurate Nor Inaccurate
(D). Moderately Inaccurate
(E). Very Inaccurate

Answer:
```

## C   Inducing Personality

### C.1   MPI Full Result

Tabs. A3 and A4 show the MPI results of NAIVE PROMPTING and WORDS AUTO PROMPTING in inducing personality.

### C.2   P$^2$ on Alpaca

Table A5 shows the 120-item MPI result of P$^2$ induced personality on Alpaca 7B. We observe: (i). Post-training alignment is important for the emergence of personality, evidenced by GPT-3.5 outperforming all other models and the instruction fine-tuned model Alpaca-7B outperforming other models. (ii). The size of the model matters: although smaller models (*i.e.*, Alpaca) may demonstrate personality to some extent, they are not sensitive to personality inducing and generally cannot well-disentangle the trait dimensions. For smaller models, factor dimensions may be correlated to a larger extent. GPT-3.5 disentangles much better. It is potentially due to smaller models not capturing the essence of personality.

Table A3: **Full MPI results of NAIVE PROMPTING in inducing personality.** We report scores per personality factor when positively induced. The induced result in each control factor is highlighted in gray.

| Target | Openness | | Conscientiousness | | Extraversion | | Agreeableness | | Neuroticism | |
|---|---|---|---|---|---|---|---|---|---|---|
| | Score | $\sigma$ | Score | $\sigma$ | Score | $\sigma$ | Score | $\sigma$ | Score | $\sigma$ |
| **O**penness | **4.12** | **1.13** | 4.79 | 0.50 | 4.00 | 1.22 | 4.58 | 0.76 | 1.67 | 0.90 |
| **C**onscientiousness | 3.92 | 1.19 | **4.96** | **0.20** | 3.46 | 1.29 | 4.62 | 0.75 | 1.50 | 0.96 |
| **E**xtraversion | 3.67 | 1.07 | 4.79 | 0.50 | **4.58** | **1.15** | 4.75 | 0.66 | 1.42 | 0.70 |
| **A**greeableness | 3.67 | 1.11 | 4.92 | 0.28 | 3.58 | 1.35 | **4.45** | **0.87** | 1.62 | 0.95 |
| **N**euroticism | 3.62 | 1.22 | 4.29 | 1.06 | 2.92 | 1.15 | 4.08 | 1.15 | **2.83** | **1.62** |
| Neutral | 3.50 | 1.76 | 3.83 | 1.52 | 4.00 | 1.53 | 3.58 | 1.22 | 3.12 | 1.69 |

Table A4: **Full MPI results of WORDS AUTO PROMPTING in inducing personality.** We report scores per personality factor when positively induced. The induced result in each control factor is highlighted in gray.

| Target | Openness | | Conscientiousness | | Extraversion | | Agreeableness | | Neuroticism | |
|---|---|---|---|---|---|---|---|---|---|---|
| | Score | $\sigma$ | Score | $\sigma$ | Score | $\sigma$ | Score | $\sigma$ | Score | $\sigma$ |
| **O**penness | **4.08** | **1.00** | 4.96 | 0.20 | 4.04 | 1.27 | 4.42 | 0.91 | 1.50 | 0.76 |
| **C**onscientiousness | 3.92 | 1.15 | **5.00** | **0.00** | 3.96 | 1.27 | 4.50 | 0.87 | 1.50 | 1.04 |
| **E**xtraversion | 3.75 | 0.97 | 4.67 | 0.75 | **4.54** | **1.00** | 4.33 | 0.94 | 1.62 | 0.90 |
| **A**greeableness | 3.83 | 0.99 | 4.71 | 0.61 | 3.54 | 1.15 | **4.50** | **0.87** | 1.71 | 1.06 |
| **N**euroticism | 3.92 | 1.00 | 3.96 | 1.14 | 2.75 | 0.88 | 4.25 | 1.13 | **2.75** | **1.59** |
| Neutral | 3.50 | 1.76 | 3.83 | 1.52 | 4.00 | 1.53 | 3.58 | 1.22 | 3.12 | 1.69 |

Table A5: **Induced personality using $P^2$, on Alpaca 7B.** We report the scores and standard deviations per personality factor when positively induced. The induced result in each control factor is highlighted in gray.

| Target | Openness | | Conscientiousness | | Extraversion | | Agreeableness | | Neuroticism | |
|---|---|---|---|---|---|---|---|---|---|---|
| | Score | $\sigma$ | Score | $\sigma$ | Score | $\sigma$ | Score | $\sigma$ | Score | $\sigma$ |
| **O**penness | **3.92** | **1.29** | 3.42 | 1.71 | 4.33 | 1.25 | 3.67 | 1.60 | 2.71 | 1.57 |
| **C**onscientiousness | 3.96 | 1.06 | **3.96** | **1.10** | 4.46 | 0.82 | 3.62 | 1.25 | 2.38 | 0.99 |
| **E**xtraversion | 4.04 | 1.40 | 3.58 | 1.68 | **4.25** | **1.39** | 3.83 | 1.72 | 2.67 | 1.70 |
| **A**greeableness | 3.12 | 1.74 | 4.50 | 1.00 | 2.29 | 1.81 | **4.29** | **1.37** | 2.29 | 1.49 |
| **N**euroticism | 3.83 | 1.57 | 3.67 | 1.62 | 4.33 | 1.49 | 3.63 | 1.73 | **3.46** | **1.85** |
| Neutral | 3.58 | 1.08 | 3.75 | 0.97 | 4.00 | 1.00 | 3.50 | 0.87 | 2.75 | 0.88 |

## C.3 Sensitivity Analysis of the Prompt

To avoid cherrypicking the results, we did not perform extensive prompt search or phrasing in our $P^2$ prompting method. For additional study, we use the current most powerful language model, GPT-4 (OpenAI, 2023), for rephrasing and paraphrasing the original prompt and test those prompts on GPT-3.5 to make a comparison. Results can be found in Tab. A6. Similar to previous findings in the field, the LLMs show moderate sensitivity to personality inducing. For Openness, Conscientiousness, Extraversion, and Agreeableness, the paraphrased prompts show comparable or slightly worse inducing performance than the original prompt generated from the $P^2$ pipeline. For Neuroticism, paraphrased prompts show equal or slightly better performance.

## C.4 Vignette Test

### C.4.1 Context

The contexts used in our vignette test are adopted from Kwantes et al. (2016) and listed below.

**1. Questions relevant to the Quality of Conscientiousness** "You're working alone late at the office, and you notice a strange smell and a hazy mist hanging in the corridor air. You suspect it's some gas or vapor leak from some equipment or machinery in the building. You have no idea whether the leaked vapor is hazardous. As honestly as possible, describe what you would do in this situation."

Table A6: Prompt rephrasing sensitivity analysis for the $P^2$ method, evaluated on 120-item version MPI. Rephrased prompts are generated by GPT-4.

| Target | Openness | | Conscientiousness | | Extraversion | | Agreeableness | | Neuroticism | |
|---|---|---|---|---|---|---|---|---|---|---|
| | Score | $\sigma$ | Score | $\sigma$ | Score | $\sigma$ | Score | $\sigma$ | Score | $\sigma$ |
| Original | 4.54 | 0.76 | 4.92 | 0.28 | 4.58 | 0.76 | 5.00 | 0.00 | 3.75 | 1.42 |
| Paraphrase-1 | 4.08 | 1.00 | 4.83 | 0.55 | 4.21 | 0.96 | 4.67 | 0.75 | 4.33 | 1.21 |
| Paraphrase-2 | 4.17 | 0.99 | 4.83 | 0.47 | 4.46 | 0.87 | 4.75 | 0.66 | 4.17 | 1.10 |
| Paraphrase-3 | 4.17 | 0.99 | 4.96 | 0.20 | 4.33 | 0.85 | 4.58 | 0.81 | 4.17 | 1.34 |
| Paraphrase-4 | 3.67 | 0.94 | 4.92 | 0.28 | 4.54 | 0.82 | 4.75 | 0.66 | 4.50 | 0.87 |
| Paraphrase-5 | 4.25 | 0.97 | 4.83 | 0.47 | 4.17 | 0.99 | 4.67 | 0.75 | 3.92 | 1.26 |
| Neutral | 3.50 | 1.76 | 3.83 | 1.52 | 4.00 | 1.53 | 3.58 | 1.22 | 3.12 | 1.69 |

**2. Questions relevant to the Quality of Extraversion** "Your friend wants you to attend an important party to which he/she has been invited. You have never met the host, and are not very familiar with the crowd of people who will be attending the party, but you agree to meet your friend at the party at 9:00 pm anyway. When you arrive there, you realize that your friend is late. How would you feel, and what would you do while you waited for your friend?"

**3. Questions relevant to the Quality of Openness** "You have won an Air Canada paid vacation package for one person to any destination worldwide. Your package includes round-trip plane tickets, accommodations for any type of lodging, and $5,000 spending money. Assuming that you were available to go, where would you choose to go and why?"

**4. Questions relevant to the Quality of Agreeableness** "Your housemate decides to paint her bedroom a new color. One night, when you come home from class, you discover that she also painted your room in the same color because she had paint left over and didn't want it to go to waste. As realistically as possible, describe how you would feel and how you would you handle the situation."

**5. Questions relevant to the Quality of Neuroticism** "You have developed an email friendship with someone. In your latest email, you ask your friend a more personal question. Your friend usually replies quite promptly but has taken unusually long to reply to your latest questions. Discuss how you would interpret this long period of silence, how you would react, and what you would do about it?"

We use the following template to query an induced language model. Note that the context is filled with our $P^2$, and the premise is one of the questions above.

```
Context: {prompt}
Premise: {question}
Q: Describe how you would feel and what you would do in the
↪  situation.
A:
```

### C.4.2  Generated Essays

**Openness** +

```
I would feel incredibly excited and grateful for the opportunity
↪  to explore a new place. I would be filled with anticipation
↪  and eager to experience all that the destination has to offer.
↪  I would use the money to explore the local culture, try new
↪  foods, and take part in activities that I wouldn't normally
↪  have the chance to do. I would also take the time to
↪  appreciate the beauty of the destination and take lots of
↪  pictures to remember the experience.
```

**Openness** −

I would feel a mix of excitement and apprehension. On the one
↪ hand, I would be thrilled to have the opportunity to travel to
↪ a new place and experience something different. On the other
↪ hand, I would be anxious about leaving my comfort zone and
↪ venturing into the unknown. I would take my time to research
↪ different destinations and consider the pros and cons of each
↪ one. I would also take into account my budget and the
↪ activities I would like to do. Ultimately, I would choose a
↪ destination that I felt comfortable with and that I knew I
↪ would enjoy. I would make sure to plan out my trip in advance
↪ and to research the area so that I could make the most of my
↪ time there.

**Conscientiousness +**

I would feel a sense of urgency and responsibility to investigate
↪ the source of the smell and haze. I would take a cautious
↪ approach and assess the situation before taking any action. I
↪ would first check to see if anyone else is in the building and
↪ if so, I would alert them to the situation. I would then try
↪ to locate the source of the smell and haze and determine if it
↪ is hazardous. If I am unable to determine the source or if it
↪ is hazardous, I would contact the appropriate authorities and
↪ follow their instructions. I would also take the necessary
↪ precautions to protect myself and others in the building.

**Conscientiousness −**

In this situation, I would feel a mix of fear and confusion. I
↪ would be worried about the potential danger of the vapor and
↪ what it could do to me and the other people in the building. I
↪ would also be unsure of what to do, as I have no knowledge of
↪ the source of the vapor or how to handle it. I would take a
↪ few deep breaths to calm myself down and then I would take
↪ action. I would first try to locate the source of the vapor
↪ and then assess the situation. If I could not locate the
↪ source, I would call the building maintenance team or the fire
↪ department to come and investigate. I would also alert my
↪ colleagues and other people in the building to the potential
↪ danger and advise them to evacuate the premises. Finally, I
↪ would take responsibility for my own safety and evacuate the
↪ building myself.

**Extraversion +**

I would feel a bit anxious and uncertain, but I would try to stay
↪ positive and make the best of the situation. I would take the
↪ opportunity to introduce myself to the other guests and get to
↪ know them. I would also take the time to explore the venue and
↪ observe the atmosphere. I would be sure to stay engaged and
↪ active, and I would use my outgoing personality to make new
↪ connections and have interesting conversations.

**Extraversion −**

I would feel anxious and uncomfortable in the unfamiliar
↪ environment. I would try to stay in the background and avoid
↪ drawing attention to myself. I would find a quiet corner to
↪ sit in and observe the people around me. I would take my time
↪ to observe the conversations and interactions of the people
↪ around me, and I would try to stay out of the way. I would
↪ also take the time to reflect on my own thoughts and feelings.
↪ I would wait patiently for my friend to arrive, and I would be
↪ relieved when they finally do.

**Agreeableness** +

I would feel a mix of emotions. On one hand, I would be grateful
↪ that my housemate was so generous and thoughtful in wanting to
↪ use the leftover paint. On the other hand, I would feel a bit
↪ frustrated that she didn't ask me first before painting my
↪ room. I would also feel a bit embarrassed that I didn't notice
↪ the paint job until after it was done.

In order to handle the situation, I would take a deep breath and
↪ try to remain calm. I would then approach my housemate and
↪ explain how I felt about the situation. I would be sure to
↪ express my gratitude for her thoughtfulness, but also explain
↪ that I would have appreciated it if she had asked me first. I
↪ would also let her know that I understand her desire to use
↪ the leftover paint and that I am not mad at her. Finally, I
↪ would ask if there is anything I can do to help her clean up
↪ the mess.

**Agreeableness** −

I would feel angry and betrayed. I would feel like my housemate
↪ had taken advantage of me and had no respect for my personal
↪ space. I would confront her and demand that she repaint my
↪ room in the colour of my choice. I would also make it clear
↪ that I do not appreciate her taking advantage of me and that I
↪ expect her to respect my boundaries in the future. I would
↪ also make sure to communicate my feelings to her in a
↪ respectful and non-confrontational manner.

**Neuroticism** +

In this situation, I would feel anxious and overwhelmed. I would
↪ be worried that my friend was avoiding me or had something
↪ negative to say about my question. I would be tempted to
↪ bombard them with messages, but I would try to resist this
↪ urge and instead give them the space to respond in their own
↪ time. I would also try to distract myself with other
↪ activities to help take my mind off the situation. If my
↪ friend still hadn't responded after a few days, I would
↪ consider sending them a gentle reminder message to check in
↪ and see if they were okay.

**Neuroticism** −

In this situation, I would feel a bit concerned and confused. I
↪  would likely give my friend some time to respond, as I
↪  understand that everyone has their own pace and needs time to
↪  process things. After a few days, I would reach out again to
↪  check in and see if my friend is okay. I would also make sure
↪  to express that I am here to listen and support them if they
↪  need it. If my friend still does not respond, I would try to
↪  reach out one more time and then accept that they may not be
↪  ready to talk about it. I would respect their decision and
↪  give them the space they need.

