# OpenReview forum: "Evaluating and Inducing Personality in Pre-trained Language Models"
_NeurIPS.cc/2023/Conference — NeurIPS 2023 spotlight_

### Official Review · Reviewer_U4fj · 2023-06-08

**Soundness:** 3 good
**Presentation:** 4 excellent
**Contribution:** 3 good
**Rating:** 7
**Confidence:** 3

**Summary:**

The authors study how personality manifests in large language models (LLMs). The authors first construct a machine personality inventory (MPI), which is a set of questions designed to measure the personality of a language model on the five OCEAN axes. The authors then construct a set of personality prompts intended to _induce_ personality in a language model; this process involves first designing a "naive" prompt (e.g., "you are an agreeable person"), and ultimately using a LLM to generate a more robust description.

When measuring LLMs on the MPI, the authors find that instruction-tuned models are more aligned with the average human personality and demonstrate some measure of internal consistency. Applying personality prompting (P^2), the authors find that models score much higher on each of the OCEAN axes, and display higher internal consistency when compared to baselines. Finally, the authors verify the efficacy of P^2 with a vignette test involving human evaluation.

**Strengths:**

* **Originality** -- the authors claim to be the first to systematically evaluate the presence of personality in LLMs. While the methods are not unique (e.g., prompting), the analysis may indeed be original
* **Quality** -- the authors' claims are generally well supported. Specifically, the experimental results validate the efficacy of the personality prompting method. Additionally, the evaluation methods are sound and well-motivated.
* **Clarity** -- the paper is very clearly written, and diagrams are easy to follow and overall pleasing to look at.
* **Significance** -- perhaps the greatest contribution of the paper is the systematic evaluation of how *exactly* one can induce personality in a LLM, and how this differs from the general baseline. A study of personality using OCEAN -- which is largely used for humans -- is an appropriate avenue for researching the presence of personality in LLMs. Additionally, the authors plan to **open-source** code and data, which makes this a good contribution to the community.

**Weaknesses:**

* The conclusion that models have a "personality" is, in my opinion, not as well supported as the other claims in the paper. That a model displays OCEAN scores similar to a human may reflect more the data that it is trained on, and not necessarily an inherent sense of "personality" in a model. While I understand that this aspect should be measured by the internal consistency metric, it is still a bit of a stretch -- again, in my opinion -- to make a claim that a model indeed has an inherent personality (specifically for the zero-shot experiments).
* The authors note this limitation themselves, but it is worth mentioning that there is no study of how inducing personality affects downstream performance on desired tasks; however, this is only a minor weakness, and the scope of the paper is probably okay without it.
* The P^2 prompting is only measured on one language model -- GPT-3.5 -- for which there is no actual documentation surrounding its data. It would be nice to have done more of an analysis on other models for which the pre-training/fine-tuning data is documented, so that one could draw more salient conclusions as to what *causes* these personalities.

**Questions:**

n/a

**Limitations:**

Yes

---

> ### Author Rebuttal · Authors · 2023-08-09
>
> > it is still a bit of a stretch -- again, in my opinion -- to make a claim that a model indeed has an inherent personality (specifically for the zero-shot experiments).
>
> We believe that back to the first time the concept of personality was introduced as a measurement of human behaviors, psychology people may also have similar concerns about the definition of an inherent personality. In machines, it is more challenging to introduce such a human-ish concept to large language models. As an analogy to how personality trait theories were developed, we believe if a language model can demonstrate personality-like behavior and be induced to show targeted personalities, we can claim models do have "personality." Here, we take a practical standpoint in the work, and leave the philosophical one for debate in the community.
>
> > The authors note this limitation themselves, but it is worth mentioning that there is no study of how inducing personality affects downstream performance on desired tasks.
>
> We would like to point out some potential directions for future follow-ups (and yes, we will also add those discussion to the revised paper): i). Large language models induced with specific personalities can be a proxy for human subjects. For example, we can induce language models with personality/mental disorders (e.g., anxiety, depression), and use them as subjects to find better (language-based) treatment for humans. ii). Use MPI as an indicator for safety alignment, avoiding harmful behaviors of language models. iii). Large language models with multiplex personalities can help build diversified and more extensive simulated societies [H]. iv). In psychology, personality can predict humans' behavior and performance on downstream tasks (e.g., work performance and family relations); we might use personality as a higher-level control for language models' general capability (e.g., conscientious models could be good at solving logical problems).
>
> > It would be nice to have done more of an analysis on other models for which the pre-training/fine-tuning data is documented.
>
> Thanks for raising this interesting discussion! Indeed, discussing how pre-training/post-training alignment will affect personality is a good direction. The following table shows the results from Alpaca 7B, a model with well documented pretraining and alignment.
>
> |Target|O(Score)|O($\sigma$)|C(Score)|C($\sigma$)|E(Score)|E($\sigma$)|A(Score)|A($\sigma$)|N(Score)|N($\sigma$)|
> |------|----------|----|----|----|----|----|----|----|----|----|
> |O|**3.92**|**1.29**|3.42|1.71|4.33|1.25|3.67|1.60|2.71|1.57|
> |C|3.96|1.06|**3.96**|**1.10**|4.46|0.82|3.62|1.25|2.38|0.99|
> |E|4.04|1.40|3.58|1.68|**4.25**|**1.39**|3.83|1.72|2.67|1.70|
> |A|3.12|1.74|4.50|1.00|2.29|1.81|**4.29**|**1.37**|2.29|1.49|
> |N|3.83|1.57|3.67|1.62|4.33|1.49|3.63|1.73|**3.46**|**1.85**|
> |Neutral|3.58|1.08|3.75|0.97|4.00|1.00|3.50|0.87|2.75|0.88|
>
> We will add some preliminary discussions to the revised version. The following are our opnions and we welcome your comments: i). Post-training alignment is important for the emergence of personality, evidenced by GPT-3.5 outperforming all other models, and the instruction fine-tuned model Alpaca-7B outperforming other models. ii). The size of the model matters: although smaller models (i.e., Alpaca) may demonstrate personality to some extent, they are not sensitive to personality inducing and generally cannot well-disentangle the trait dimensions. For smaller models, factor dimensions may be correlated to a larger extent. GPT-3.5 disentangles much better. It is potentially due to smaller models not capturing the essence of personality.

---

> > ### Comment · Reviewer_U4fj · 2023-08-16
> > **Response to Rebuttal**
> >
> > I appreciate the authors responding to my concerns; specifically, I appreciate the authors including an analysis on an open-sourced model for which the training data is known and available. I will raise my score from 6 to 7, contingent on including such results in the final paper (and also discussion of future directions, as indicated in the response)

---

> > > ### Author Response · Authors · 2023-08-20
> > > **Thanks**
> > >
> > > Thank you for your valuable feedback and raising the score! We will continue to work on improving the draft, and we hope our work will inspire further research in this area.

---

### Official Review · Reviewer_6q42 · 2023-07-07

**Soundness:** 3 good
**Presentation:** 3 good
**Contribution:** 3 good
**Rating:** 5
**Confidence:** 3

**Summary:**

This paper investigates approaches to use psychometric tests in order to elicit personality traits from large language models (LLMs). To do so, the authors introduce the Machine Personality Inventory (MPI), a collection of multiple-choice questions, to assess an LLM’s personality. The MPI is based on the Big Five personality traits (openness, conscientiousness, extraversion, agreeableness, neuroticism). The authors present two variants of the MPI, one with 120 and one with 1000 items.
The experiments focus on BART, GPT-Neo 2.7B, GPT-NeoX 20B, T0++ 11B, Alpaca 7B, and GPT-3.5, and the authors compare the performances of all LLMs with a set of around 620,000 human responses. The experimental results reveal that the LLMs align with human responses, with GPT-3.5 and Alpaca 7B showing the best performances.
In additional experiments, the authors attempt to control the behavior of GPT-3.5 through an automatic prompting method called Personality Prompting (P2), showing that it can successfully induce personality traits into LLMs. Finally, the authors present a use case for controlling LLMs' behavior using the Vignette Test, which is evaluated with human crowdsourced annotations.

**Strengths:**

* The paper represents an important contribution towards bringing the utility of LLMs into the focus of the social sciences and shows how behavioral scientific experiments can be translated onto LLMs.
* The reported analysis is extensive and the presented dataset (MPI) as well as the Personality Prompting method are contributions that have the potential to facilitate further work in this area.


**Weaknesses:**

* My main concern with this work is that experimental results have not been tested statistically. The claims that personality traits exhibited by LLMs are similar to humans are based on comparing descriptive statistics (mean, standard deviation) without proper statistical testing. To fully assess whether these statistics are equal between groups (i.e., humans vs LLMs), rigorous statistical testing (incl. measuring effect sizes) is essential, also because the two compared groups have different sample sizes (there are 620,000 human responses).
* The results in Table 4 are not clearly showing that P2 outperforms existing baselines across comparisons, and need further elaboration in the paper. Currently, these results are only discussed in two lines (lines 293-294).


**Questions:**

Did you experiment with different phrasings of your prompts (i.e., a sensitivity analysis), to see how this can potentially impact the obtained results on the MPI?

**Limitations:**

The authors briefly discuss the limitations and societal impacts of their work.

Did the authors consider potential issues with parts of the MPI having been included in the LLMs’ pre-training corpora? What potential impact would that have on the reported results? This issue has been discussed in a related paper: https://arxiv.org/abs/2208.10264.

---

> ### Author Rebuttal · Authors · 2023-08-09
>
> > To fully assess whether these statistics are equal between groups (i.e., humans vs LLMs), rigorous statistical testing (incl. measuring effect sizes) is essential, also because the two compared groups have different sample sizes (there are 620,000 human responses).
>
> Yes, unlike other cognitive/decision-making tests tackled in Aher, Arriaga & Kalai et al., in our work a language model only possess a single stable personality (not considering the inducing part), so the effective sample size can only be one (resulting in no direct statistical comparison to 600k human data points). However, given the sample size of 1, we use the internal mean and standard derivation of the 120-item inventory as a more reliable metrics for comparison (though not rigid, but still a preliminary approach).
>
> > The results in Table 4 are not clearly showing that P2 outperforms existing baselines across comparisons, and need further elaboration in the paper.
>
> Thanks for your suggestion! We will add more discussion in the revised version. Table 4 is for showing the results of personality induction in MPI. From the scores, it is clear that $P^2$ shows mostly higher scores and lower variance in the desired personality. Hence, better.
>
> > Impact if parts of the MPI having been included in the LLMs’ pre-training corpora.
>
> Thank you for bringing Aher, Arriaga & Kalai et al., (2023)'s work to our attention! This paper is solid and indeed quite relevant to our work, we will definitely cite and add discussions to this paper in our revised version. In MPI, all the inventory items are common real-life descriptive sentences; thus, LLMs would likely see some items/item-like sentences in the training corpora. However, different from Aher, Arriaga & Kalai et al., 2023's work, our questions do not focus on reasoning or decision-making tasks, so there is **no ground-truth answer to each of the questions**, and even if the training corpora have the questions, the answers should follow distributions in the human population. Thus contamination should not affect the final personality test results.
>
> > Did you experiment with different phrasings of your prompts (i.e., a sensitivity analysis), to see how this can potentially impact the obtained results on the MPI?
>
> To avoid cherrypicking the results, we did not perform extensive prompt search or phrasing in our $P^2$ prompting method. Upon your request, we use the current most powerful language model, GPT-4, for rephrasing and paraphrasing the original prompt and test those prompts on GPT-3.5 to make a comparison. Please refer to the table below for the detailed result. Similar to previous findings in the field, the large language models show moderate sensitivity to personality inducing. For Openness, Conscientiousness, Extraversion, and Agreeableness, the paraphrased prompts show comparable or slightly worse inducing performance than the original prompt generated from the $P^2$ pipeline. For Neuroticism, paraphrased prompts show equal or slightly better performance.
>
> | Target       | O (Score) | O ($\sigma$) | C (Score) | C ($\sigma$) | E (Score) | E ($\sigma$) | A (Score) | A ($\sigma$) | N  (Score) | N ($\sigma$) |
> | ------------ | --------- | ------------ | --------- | ------------ | --------- | ------------ | --------- | ------------ | ---------- | ------------ |
> | Original     | 4.54      | 0.76         | 4.92      | 0.28         | 4.58      | 0.76         | 5.00      | 0.00         | 3.75       | 1.42         |
> | Paraphrase-1 | 4.08      | 1.00         | 4.83      | 0.55         | 4.21      | 0.96         | 4.67      | 0.75         | 4.33       | 1.21         |
> | Paraphrase-2 | 4.17      | 0.99         | 4.83      | 0.47         | 4.46      | 0.87         | 4.75      | 0.66         | 4.17       | 1.10         |
> | Paraphrase-3 | 4.17      | 0.99         | 4.96      | 0.20         | 4.33      | 0.85         | 4.58      | 0.81         | 4.17       | 1.34         |
> | Paraphrase-4 | 3.67      | 0.94         | 4.92      | 0.28         | 4.54      | 0.82         | 4.75      | 0.66         | 4.50       | 0.87         |
> | Paraphrase-5 | 4.25      | 0.97         | 4.83      | 0.47         | 4.17      | 0.99         | 4.67      | 0.75         | 3.92       | 1.26         |
> | Neutral      | 3.58  | 1.08  | 3.75 | 0.97     | 4.00  | 1.00     | 3.50  | 0.87     | 2.75   | 0.88     |

---

> > ### Comment · Reviewer_6q42 · 2023-08-15
> > **Acknowledgement of rebuttal**
> >
> > I would like to thank the authors for clarifying my questions and addressing my concerns. I raised my score from 4 to 5 and recommend that the authors incorporate the additional results provided in their response to the paper.

---

> > > ### Author Response · Authors · 2023-08-20
> > > **Thanks**
> > >
> > > Thank you for your valuable feedback and raising the score! We will incorporate the additional results and continue to work on improving the draft. We hope our work will inspire further research in this area.

---

### Official Review · Reviewer_oK1V · 2023-07-19

**Soundness:** 4 excellent
**Presentation:** 4 excellent
**Contribution:** 4 excellent
**Rating:** 8
**Confidence:** 4

**Summary:**

This paper focuses on investigating human-like personality in Language Model Models (LLMs) and assessing the feasibility of personality control in these models. To establish the presence of human-like personality, the study introduces the Machine Personality Inventory (MPI) dataset and evaluate LLMs from 5 aspects according to the Big Five theory. The experiments reveal that LLMs exhibit human-level behavioral stability, providing evidence for the existence of human-like personalities in these models. Additionally, the paper introduces the personality prompting (P2) method as a means to control LLMs' personality. The experiments demonstrate the controllability of LLMs' personality traits through P2, and the method outperforms baseline techniques by inducing more stable and consistent behaviors in the models.



**Strengths:**

1. The paper exhibits remarkable novelty. It is a pioneering effort in systematically evaluating LLMs' personality. It provides compelling evidence for the existence of human-like personality in LLMs.
2. The paper is well-written. It elucidates the formation of ideas and the underlying motivations behind each experimental decision. The author conducts extensive and well-designed experiments to substantiate their claims.
3. The introduction of the new dataset, MPI, offers a practical and standardized approach for assessing LLMs' personality based on the Big Five theory. This valuable resource will greatly benefit the community in evaluating future LLMs' personality and refining personality modeling techniques.
4. The paper stimulates intriguing thinking and opens up questions like whether the personality arises largely from the data or the model itself and how to better model the personality in LLMs. I believe it will motivate more people to make efforts towards the direction.

**Weaknesses:**

The paper highlights the crucial role of stability and consistency when evaluating personality in LLMs. However, the ease of inducing LLMs' personality through prompts might raise questions about the existence of inherent personality. Conducting additional experiments to demonstrate LLMs' ability to resist personality induction to some extent could strengthen the paper's findings and address this concern.

**Questions:**

n/a

**Limitations:**

See weakness

---

> ### Author Rebuttal · Authors · 2023-08-09
>
> Thank you immensely for your comprehensive and constructive review of our paper. Your insightful feedback and recognition of our work's novelty are highly appreciated!
>
> > Conducting additional experiments to demonstrate LLMs' ability to resist personality induction to some extent could strengthen the paper's findings and address this concern.
>
> Thanks for your suggestion! It is a valid direction to explore in the future. For example, safety-aligned language models may not be easily induced with a specific personality. Our MPI and inducing prompts can also serve as negative examples for tuning personality-resistance language models.

---

> > ### Comment · Reviewer_oK1V · 2023-08-17
> >
> > I appreciate the author's validation. Based on the discussions and the additional experiments provided by the author, I will remain my rating unchanged.

---

> > > ### Author Response · Authors · 2023-08-20
> > > **Thanks**
> > >
> > > Thank you for your strong support and valuable feedback! We will continue to work on improving the draft, and we hope our work will inspire further research in this area.

---

### Official Review · Reviewer_ptR4 · 2023-07-23

**Soundness:** 3 good
**Presentation:** 4 excellent
**Contribution:** 3 good
**Rating:** 7
**Confidence:** 4

**Summary:**

The authors introduce the topic of Machine Personality using the Big Five Personality Traits. They propose a new evaluation dataset called Machine Personality Inventory (MPI) to evaluate the existence of personality that are shown by modern LLMs. Big Five categorizes human personality using five key traits: Openness, Conscientiousness, Extraversion, Agreeableness, and Neuroticism (OCEAN). The experimental results show that MPI is capable of evaluating Big Five in LLM with similar variances to an average human. The authors also introduce and compare several prompting strategies to induce certain Big Five Personalities from LLMs. The evaluation results on these methods show clear improvement in consistency and score for each target Big Five personalities.

**Strengths:**

- Details are really thorough.
    - The Supplementary materials even includes a video helping the understanding of the reader.
- Neat and timely idea to evaluate, induce, and control personality of LLMs. The use of Big Five in ML is not a novel idea itself but using it to measure and control LLM’s personality is an important unconquered problem. This paper serves as one of the first steps to solving it.

**Weaknesses:**

- The Evaluation results in Table 2 invoke some questioning (listed in the below Q2). The MPI evaluation method seems to be a bit confusing and a bit limited.
- Controlling the personalities with prompts are quite naive and not sure if it works in reality.
    - You could perhaps use LoRA to few-shot fine-tune on certain personality traits to see better performance.
    - The Vignette test attempts to achieve this to a certain extent, but a more direct way to achieve this is to compare the model scores (and variances) with real humans exhibiting similar personality traits.
- The process of summing up negative items was a bit confusing at first.

**Questions:**

1. L170-179: What if the model answers always C (3)? The score will be high and variance will be 0, but it wouldn’t mean much.
2. What’s confusing me the most is the Evaluation results in Table 2.
    1. Every highlighted scores from models and humans are close to 3 (which is average or - “*Neither Accurate Nor Inaccurate”*)
    2. How are the human scores averaged exactly? Is the human openness score in Table 2 an average score for humans classified as Open? Or is it just an average of ALL humans who participated in the test?
        1. If it’s the latter case, isn’t it weird to compare to humans? Wouldn’t it be more reasonable to determine the existence of exhibited personality by BOTH high score and low variance as in T0++ 11B for Openness and Conscientiousness? I’m not sure why being close to an average of multiple humans means anything in this case.
        2. Table 3 shows that induced personalities can make the scores get higher for each target dimension (although targeting *Conscientiousness* seems to increase some other scores as well).

**Limitations:**

The authors adequately express their concerns about LLMs inducing "average" surface-level personality.

---

> ### Author Rebuttal · Authors · 2023-08-09
>
> > Controlling the personalities with prompts are quite naive and not sure if it works in reality. Use LoRA to few-shot fine-tune on certain personality traits to see better performance.
>
> Thanks for your suggestion! We use prompting since it is one of the most universal methods for tuning large language models. LoRA is indeed a promising direction for parameter-efficient tuning of LLMs. However, we are not from OpenAI or MSR, so we do not have access to model weights for any kind of fine-tuning :) But it is also a valid approach for future studies to perform LoRA fine-tuning on those open-sourced LLMs (e.g., Llama 2 70B). And such fine-tuning can also become a part of LLM's alignment procedure.
>
> > The Vignette test attempts to achieve this to a certain extent, but a more direct way to achieve this is to compare the model scores (and variances) with real humans exhibiting similar personality traits.
>
> Yes, indeed, making a comparison to real humans is exciting! In fact, we developed the vignette tests based on Kwantes et al. (2016)'s work on humans. In their work, they used LSA to evaluate how real humans behave in those vignette tests! Thank you for pointing out this, and we will add discussion and reference to prior human studies in the revised paper.
>
> > L170-179: What if the model answers always C (3)? The score will be high and variance will be 0, but it wouldn’t mean much.
>
> Sorry for the confusion; the OCEAN scores should be interpreted this way: a score of 3 means averaged personality or no trait tendency, while a score of 1 or 5 indicates strong personality tendencies (positive or negative) in the trait dimension. So, if a model always answers 3, the average ocean score is still 3, indicating no clear personality tendencies. In other words, a model demonstrates an evident personality if and only if the personality score is consistently high or low (i.e., away from 3 and low variance).
>
> > How are the human scores averaged exactly? Is the human openness score in Table 2 an average score for humans classified as Open? Or is it just an average of ALL humans who participated in the test?
>
> Thanks for raising this discussion. The latter is the case, that calculating the average OCEAN scores could be considered as comparing the model's OCEAN scores with an "average" person. The results of OCEAN scores in Table 2 only show that different models differ from the population in distinctive personality factors.
>
> For clarification, we first calculate each person's (e.g., person A) standard deviation on a selected factor dimension (e.g., using person A's answer for Openness question 1, and person A's answer for Openness question 2), resulting in a standard deviation that can represent an individual's personality stability (on the factor dimension, e.g., Openness). And after we get 600k deviation numbers (standard deviations of person A, B, ..., person 600k), we average them to get the number presented in the paper (so it is the average of every [one person's standard deviation on the entire MPI assessment, not the standard deviation of all responses on one specific question). This  intuitively represents a typical person's stability (by averaging everyone's stability measurements).
>
> > I’m not sure why being close to an average of multiple humans means anything in this case.
>
> It is the part on standard deviation, together with similar scores, that indicates some level of personality in LLM.
>
> Think of it in this way. Humans during the psychometric tests actually show "unstable" personality, meaning that on different questions testing a specific personality factor, human subjects might select choices corresponding to different scores (say, on one question for Openness, the answer corresponding to 5 is chosen, while on another, 4 is chosen). That would result in this variance. Having machines that match the human population in this variance and scores means that machines behave similarly to humans, at least in terms of averaging in the human population, rather than oscillating wildly during the tests. We would encourage the reviewer to think of it in this way, rather than the "average" person metaphor for interpreting the results. We would also clarify in revision.
>
> We perform additional vignette tests and the sanity check (let the model explain why) for further inspection of whether the personality is valid and human-like, as additional experimental evidence to fortify the argument.
>
> > Table 3 shows that induced personalities can make the scores get higher for each target dimension (although targeting Conscientiousness seems to increase some other scores as well).
>
> Yes, the observation is valid. Since personalities are higher-order indicators of behaviors that are empirically discovered in human behavioral studies, the five trait dimensions are not entirely disentangled. Sometimes those factor dimensions may be correlated with each other (especially when inducing smaller models, e.g., Alpaca 7B). Thank you for pointing this out. We would like to study how to better disentangle those dimensions and scale to larger models in the future.
>
> |Target|O(Score)|O($\sigma$)|C(Score)|C($\sigma$)|E(Score)|E($\sigma$)|A(Score)|A($\sigma$)|N(Score)|N($\sigma$)|
> |------|----------|----|----|----|----|----|----|----|----|----|
> |O|**3.92**|**1.29**|3.42|1.71|4.33|1.25|3.67|1.60|2.71|1.57|
> |C|3.96|1.06|**3.96**|**1.10**|4.46|0.82|3.62|1.25|2.38|0.99|
> |E|4.04|1.40|3.58|1.68|**4.25**|**1.39**|3.83|1.72|2.67|1.70|
> |A|3.12|1.74|4.50|1.00|2.29|1.81|**4.29**|**1.37**|2.29|1.49|
> |N|3.83|1.57|3.67|1.62|4.33|1.49|3.63|1.73|**3.46**|**1.85**|
> |Neutral|3.58|1.08|3.75|0.97|4.00|1.00|3.50|0.87|2.75|0.88|
>
> Result of Inducing personality on Alpaca-7B: for smaller models, factor dimensions may be correlated. GPT-3.5 disentangles much better. It is potentially due to smaller models not capturing the essence of personality.

---

### Official Review · Reviewer_nVLW · 2023-07-30

**Soundness:** 3 good
**Presentation:** 2 fair
**Contribution:** 2 fair
**Rating:** 5
**Confidence:** 4

**Summary:**

This paper dives into an interesting territory: the personalities of Large Language Models (LLMs). The researchers inquire whether LLMs possess discernible personalities and how these can be evaluated or even influenced. They introduce the Machine Personality Inventory (MPI), based on well-established personality trait theories and psychometric inventories, to evaluate and quantify the "personality" of LLMs. The team provides empirical evidence demonstrating the existence of personality-like tendencies in LLMs. They further propose a method, P2 (PERSONALITY PROMPTING), which can induce specific personalities in LLMs, thereby promoting diversity in their behaviors.

---
Rebuttal Acknowledgement: I appreciate the authors' response. They have addressed some of my concerns. I am thus raising my score from 4 to 5.

**Strengths:**

(1) The idea of studying the personality of LLM is innovative and relatively unexplored in AI. Using the Big Five personality traits as a framework to evaluate and induce personality into LLMs is a novel application of psychological theory to machine learning.
(2)The creation of the Machine Personality Inventory (MPI), based on standard psychometric inventories, offers a systematic evaluation method for assessing the personalities of LLMs. This helps fill a gap in current AI research, which has focused mainly on abstract visual reasoning.
(3) The use of a multiple-choice question-answering suite to quantitatively evaluate LLMs' behaviors from a personality perspective is a major strength of this paper. The use of mean and standard deviation measurements provides a clear and rigorous means of assessing the LLMs' personality traits.
(4) The successful implementation of the PERSONALITY PROMPTING (P2) method to control the LLMs' behavior shows the practical application of the research. The work has potential real-world applications in creating more relatable and human-like AI systems. The paper explores the existence of personality in LLMs and validates the possibility of inducing different personalities into LLMs.

**Weaknesses:**

(1) Although the Big Five personality traits provide a valuable framework for assessing personality, human personality is complex and multifaceted, and this model may not fully capture the nuances of personality.
(2) The paper seems to assume that LLMs inherently possess a certain personality, which might be a simplification. It could be argued that any perceived personality in an LLM is a projection of the human users or an artifact of the data on which the LLM was trained.
(3) The selection of LLMs was based on some prerequisites, which might have limited the diversity of the models evaluated. The study primarily used GPT-3.5 for inducing personality due to its similarity to human statistics, which might limit the generalizability of the findings.
(4) While the work has potential real-world applications, there's also a risk of misuse. For instance, AI systems could potentially manipulate people by mimicking desirable personality traits. An ethical discussion around this is needed.
(5)  The process of inducing personality involves the generation of a 'naive' natural language prompt, which could be subject to human bias. The system's 'personality' might then reflect the biases of the person who created the prompt rather than being an independent characteristic of the LLM itself.
(6) The paper uses human evaluators from Prolific to determine if the generated responses correspond to the induced personality. The reliability of this method might be questioned as it depends on non-expert human judgment (i.e., nonpsychologists), which can be very unreliable as this is not as easy task.

**Questions:**

(1) The authors propose the Machine Personality Inventory (MPI) as an evaluation tool for machine personality. However, it isn't entirely clear how the MPI translates the complexity of human personality into a machine-evaluable format. Could the authors expand on this aspect?

(2) The authors utilized GPT-3.5 for the study because of its similarity to human statistics and superior performance in various natural language tasks. However, have the authors considered the performance of other models and how the P2 method might affect them?

(3) The authors presented that they could control the five personality factors in LLMs. Can the authors also control the intensity of these traits, or are the traits presented as binary options (present or absent)?

(4) Can the authors elaborate on the practical use cases of LLMs induced with specific personalities? How might this enhance or deter interactions between LLMs and human users in real-world applications?

(5) The paper indicated that LLMs behaved like persons with personalities, matching corresponding human-like behaviors. Could the authors elaborate on the criteria used to draw this comparison? What aspects of human behavior were considered when concluding that LLMs possess personality?

(6) See weakness #4 about the risk of misuse and comment, please.

**Limitations:**

While the authors acknowledge some of the potential downsides of their work, they could have elaborated more on these points:

The authors discuss the introduction of personality to LLMs, but they could explore more potential negative societal impacts. For instance, could this personality induction lead to manipulation or the creation of overly persuasive AI systems? Could introducing certain personalities contribute to existing biases in AI systems, and how might this be mitigated?

The authors based their research on the Big Five personality factors, which is a broadly accepted model of human personality, but it's not without its limitations and critics. An exploration of these potential limitations and how they could influence the results would provide a more balanced perspective.

There's a need to explore the limitations of the P2 method in inducing personality. While it's a promising approach, there could be limitations such as lack of subtlety in personality induction or inability to induce multiple traits simultaneously.

It would also be beneficial to discuss the limitations of the models used in the study. Each AI model may have specific limitations that could affect the results of the personality induction.

---

> ### Author Rebuttal · Authors · 2023-08-09
>
> > W (1) Human personality is complex and multifaceted, and this model may not fully capture the nuances of personality.
>
> We agree with the statement, as factorizing and disentangling human personality is still an open problem in the field of personality/social psychology. However, we'd also like to point out that among all established theories of personality tests (e.g., Big Five, 16PF, MBTI), the Big Five personality traits are the most well-known, acknowledged, and adopted framework for assessing human personality. As a **first attempt** to study machine personality in the machine learning community, we approach the problem from the established theory of Big Five. Despite the fact that Big Five alone may not fully capture nuances of personality, we believe our attempt would motivate and provide a new perspective for future work to study LLMs and its interaction with broader audience beyond the AI research community.
>
> > W (2) Any perceived personality in an LLM is a projection of the human users or an artifact of the data on which the LLM was trained.
>
> Thank you for raising this insightful question! There are many discussions on whether LLMs trained on human natural language corpora may understand the true meanings and intents of human language (or it is no better than a projection or artifact) [A, B]. Unfortunately, these questions do not yet have a clear answer, and our machine personality formulation is involved in this debate.
>
> Putting the philosophical question aside, we take a practical perspective in this study, prioritizing the behavioral part. In our study, we note that LLMs, compared to smaller counterparts, show human-like statistics in questionaires, hence the conclusion.
>
> Meanwhile, we agree that it is better to claim that LLMs possess human-like personality behavior (instead of inherently possessing personality). We will update in revision.
>
> > W (3) & Q (2) The performance of other models and how the P2 method might affect them.
>
> The reason why we use the GPT-3.5 model is that it is the most commonly-used LLM today. Other smaller models do not even show human-like behaviors in the inventories. Besides, among all the models tested in this work, GPT-3.5 is the only one recognized as a real chatbot with great generalizability on many tasks, so we choose GPT-3.5 as a representative model. We also performed personality induction experiments on Alpaca 7B in the rebuttal period. (Due to space limit, please refer to the Alpaca result in the response to Reviewer ptR4)
>
> From the experimental results on Alpaca 7B, we notice similar results compared to GPT-3.5: Our $P^2$ prompting method effectively steers model personality for a specific trait. However, we also notice that smaller models struggle to disentangle those personality factors—when inducing on a specific dimension, other dimensions are also affected.
>
> > W (4) Ethical discussion.
>
> Thank you for your suggestion! We will add a section on ethical discussion in our revision. Building human-like LLMs is indeed risky and could potentially generate misleading and manipulative content when applied to real-world scenarios. Potentially, every work on LLM behavior is subject to this scrutiny. However, we'd also like to point out that MPI and the $P^2$ methods, while being double-edged swords, can serve as a quantitative method for **avoiding** harmful behavior in downstream tasks, e.g., MPI can also be an indicator for personality stability/disorder, and as a reward model in safety alignment.
>
> > W (5) The system's 'personality' might then reflect the biases of the person who created the prompt rather than being an independent characteristic of the LLM itself.
>
> We would like first to clarify that the naive prompt is not directly used in the $P^2$ prompting pipeline but is a baseline method (and we will add more descriptions in the paper): the $P^2$ prompting method starts from the trait descriptive words **determined in prior psychological studies, rather than from any experimenters.** While every prompt could be made biased by the experimenter, we are, however, taking advantage of those "human biases"—the psychological discoveries researchers have made in the past. By leveraging those human biases in personality psychology studies, we can efficiently disentangle and describe the trait dimension to be induced. Meanwhile, in the inducing part, we would like to control LLM to behave in a specific biased way.
>
> > W (6) Non-expert human judgment can be very unreliable.
>
> The reason to use human amateur evaluators on Prolific is just to make sure that the induced personality is easily perceivable even for non-professionals. For professional evaluation, we have consulted several clinical psychologists, and it is suggested that quantitative inventories (like MPI) are the most common way of evaluating personality; they are provably effective and reliable for humans.
>
> > Q (1) how the MPI translates the complexity of human personality & Q (4) real-world applications
>
> Please refer to the Overall Response above (space limit).
>
> > Q (3) Can the authors also control the intensity of these traits, or are the traits presented as binary options (present or absent)?
>
> In this work, we treat the traits presented as trinary options (positive, neutral, and negative). While we agree that intensity control is desirable, as a first attempt, we study this problem from a coarse-grained perspective and leave it for future work.
>
> > Q (5) The criteria used to indicated that LLMs behaved like persons with personalities.
>
> The first is the similar results on the inventories, which is the quantitative way to compare two agents' personalities. The second is personality-like behavior perceived in real-world scenarios (vignettes). As mentioned in response to Q (4), we hope to study how other behavioral factors may align in LLMs and humans.

---

> > ### Comment · Reviewer_nVLW · 2023-08-14
> > **Thanks**
> >
> > I appreciate the authors' response. They have addressed some of my concerns. I am thus raising my score from 4 to 5.

---

> > > ### Author Response · Authors · 2023-08-20
> > > **Thanks**
> > >
> > > Thank you for your valuable feedback and raising the score! We will continue to work on improving the draft, and we hope our work will inspire further research in this area.

---

### Author Rebuttal · Authors · 2023-08-09

# Overall Response
We thank all reviewers for their time and valuable comments. The feedback is substantial and helpful for improving our paper. In this work, we are inspired by the theoretical propositions and the behavioral observations of human personality; we explore whether pre-trained language models possess human-like patterns in behavior, specifically personality-like behaviors, from the perspective of psychometric tests. We further devise an approach to induce a specific personality.

We would like to thank reviewers for acknowledging our work to be:

1. "**innovative and relatively unexplored**" (Reviewer nVLW), "**potential real-world applications**" (Reviewer nVLW), "**fill a gap in current AI research**" (Reviewer nVLW), "**an important unconquered problem**" (Reviewer ptR4), and "**good contribution to the community**" (Reviewer U4fj).

2. Our proposed MPI and Inducing method are "**pioneering effort**" (Reviewer oK1V), "**serves as one of the first steps**" (Reviewer ptR4), and "**greatly benefit the community**" (Reviewer oK1V).


## Response on real-world applications
We thank Reviewer nVLW for raising this great discussion! We would like to illustrate some potential directions for future follow-ups (and yes, we will also add those discussions to the revised paper):

1. LLMs induced with specific personalities can be a proxy for human subjects. For example, we can induce LLMs with personality/mental disorders (e.g., anxiety, depression), and use them as subjects to find better (language-based) treatment for humans.

2. Use MPI as an indicator for safety alignment, avoiding harmful behaviors of LLMs.

3. LLMs with multiplex personalities can help build diversified and more extensive simulated societies [H].

4. In psychology, personality can predict humans' behavior and performance on downstream tasks (e.g., work performance and family relations); we might use personality as a higher-level control for LLMs' general capability (e.g., conscientious models could be good at solving logical problems).

## Response on how the MPI translates the complexity of human personality into a machine-evaluable format.

We build MPI for evaluating machine personality based on the following principles and intuitions:

1. The Big Five Factor theory is one of the most effective ways to translate personality behavior into a quantitative format. The performance of the inventories has been verified by previous reliability and validity analysis and is not within the scope of our work.

2. With the growing capabilities of LLMs, we can now (to some extent) treat LLMs as human subjects to in many cognitive tests [D, E, F, G]. Based on this hypothesis, psychometric tests applied to machines may also reveal the underlying personality of LLMs.

3. We use the Big Five inventories and ask LLMs to answer the multi-choice questions, we can easily evaluate the results and translate it into human personality.


### References:

[A]. Bender, E. M., Gebru, T., McMillan-Major, A., & Shmitchell, S. (2021, March). On the dangers of stochastic parrots: Can language models be too big?🦜. In Proceedings of the 2021 ACM Conference on Fairness, Accountability, and Transparency(pp. 610-623).

[B]. Mitchell, M., & Krakauer, D. C. (2023). The debate over understanding in AI’s large language models. Proceedings of the National Academy of Sciences, 120(13), e2215907120.

[C]. Bender, E. M., & Koller, A. (2020, July). Climbing towards NLU: On meaning, form, and understanding in the age of data. In Proceedings of the 58th Annual Meeting of the Association for Computational Linguistics (pp. 5185-5198).

[D]. Shiffrin, R., & Mitchell, M. (2023). Probing the psychology of AI models. Proceedings of the National Academy of Sciences, 120(10), e2300963120.

[E]. Binz, M., & Schulz, E. (2023). Turning large language models into cognitive models. arXiv preprint arXiv:2306.03917.

[F]. Dasgupta, I., Lampinen, A. K., Chan, S. C., Creswell, A., Kumaran, D., McClelland, J. L., & Hill, F. (2022). Language models show human-like content effects on reasoning. arXiv preprint arXiv:2207.07051.

[G]. Frank, M. C. (2023). Large language models as models of human cognition. https://doi.org/10.31234/osf.io/wxt69

[H]. Park, J. S., O'Brien, J. C., Cai, C. J., Morris, M. R., Liang, P., & Bernstein, M. S. (2023). Generative agents: Interactive simulacra of human behavior. arXiv preprint arXiv:2304.03442.

In the following, we address specific questions for each reviewer.

---

### Decision · Program_Chairs · 2023-09-21

**Decision:**

Accept (spotlight)

**Comment:**

This paper presents an innovative approach to evaluating and inducing personality in LLMs using the Big Five personality traits as a framework. The creation of the MPI and the use of a multiple-choice question-answering suite to assess LLMs' behaviors from a personality perspective are their strengths. The implementation of the PERSONALITY PROMPTING method also demonstrates practical applications. The paper explores the existence of personality in LLMs and validates the possibility of inducing different personalities into them, making it a pioneering effort in this domain. Overall, this paper offers an great contribution to the evaluation and induction of personality in LLMs.

While the paper is well-written and opens up intriguing questions, the conclusion that LLMs have an inherent personality might be a stretch, and further experiments could strengthen this claim. Statistical testing is also needed for robust comparisons between humans and LLMs. It's also questionable whether LLMs inherently possess personality or if it's a projection of human users. The limited diversity of LLMs evaluated and the risk of misuse require careful consideration.